# Retinal orientation and interactions in rhodopsin reveal a two-stage trigger mechanism for activation

Naoki Kimata[1,*], Andreyah Pope[1,*], Markus Eilers[1], Chikwado A. Opefi[2], Martine Ziliox[1], Amiram Hirshfeld[3], Ekaterina Zaitseva[4], Reiner Vogel[5], Mordechai Sheves[3], Philip J. Reeves[2] & Steven O. Smith[1]

The 11-*cis* retinal chromophore is tightly packed within the interior of the visual receptor rhodopsin and isomerizes to the all-*trans* configuration following absorption of light. The mechanism by which this isomerization event drives the outward rotation of transmembrane helix H6, a hallmark of activated G protein-coupled receptors, is not well established. To address this question, we use solid-state NMR and FTIR spectroscopy to define the orientation and interactions of the retinal chromophore in the active metarhodopsin II intermediate. Here we show that isomerization of the 11-*cis* retinal chromophore generates strong steric interactions between its β-ionone ring and transmembrane helices H5 and H6, while deprotonation of its protonated Schiff's base triggers the rearrangement of the hydrogen-bonding network involving residues on H6 and within the second extracellular loop. We integrate these observations with previous structural and functional studies to propose a two-stage mechanism for rhodopsin activation.

[1] Department of Biochemistry and Cell Biology, Stony Brook University, Stony Brook, New York 11794-5215, USA. [2] School of Biological Sciences, University of Essex, Wivenhoe Park, Essex CO4 3SQ, UK. [3] Department of Organic Chemistry, Weizmann Institute, Rehovot 76100 Israel. [4] University of Freiburg, Institute of Physiology II, D79104 Freiburg Germany. [5] Biophysics Section, Institute of Molecular Medicine and Cell Research, Albert-Ludwigs-University, D-79104 Freiburg, Germany. * These authors contributed equally to this work. Correspondence and requests for materials should be addressed to S.O.S. (email: steven.o.smith@stonybrook.edu).

The visual receptor rhodopsin is a member of the family A G protein-coupled receptors (GPCRs)[1,2]. These receptors share a seven transmembrane (TM) helix architecture and the ability to activate heterotrimeric G proteins, yet they respond to a wide array of ligands ranging from small-molecule odorants in the olfactory receptors to peptide ligands in the hormone and chemokine receptors[1,2]. The diversity between receptor subfamilies largely lies in the extracellular, ligand-binding region, which has evolved to recognize and respond to different types of signals[3,4]. The extracellular loops and extracellular ends of the TM helices contain many subfamily-specific residues, while most of the sites with high sequence conservation across the family A GPCRs are found in the TM core and intracellular G protein-binding cavity[1,5]. A common feature of GPCR activation is the outward rotation of the intracellular end of TM helix H6, which serves to expose the G protein-binding site[6–8]. Nevertheless, the molecular mechanism by which this intracellular motion is achieved on extracellular binding of such diverse ligands remains largely unresolved.

Rhodopsin provides an ideal model system for addressing the activation mechanism of family A GPCRs. Its light-sensitive retinal chromophore is covalently bound via a protonated Schiff's base (PSB) linkage to Lys296[7.43] (superscripts denote generic Ballesteros–Weinstein numbering of GPCRs[9]) in the interior of the protein. This covalent attachment ensures full ligand occupancy, a desirable property for structural studies. The retinal-binding site has evolved to accommodate the 11-*cis* isomer of the retinal PSB[10], which acts as an inverse agonist and locks the receptor in a fully off-state. Photochemical isomerization and deprotonation of the PSB forms an all-*trans* retinal SB chromophore, which acts as a full agonist for receptor activation. That is, rhodopsin functions as a molecular off–on switch; it is designed to be fully inactive in the dark and to rapidly convert to a fully active structure in the light. This activation process differs from GPCRs that bind diffusible agonists where the signalling status is more complex and governed by an equilibrium between ligand-bound and ligand-free states[11].

The crystal structure of the apoprotein opsin[6,7] and constitutively active mutants of rhodopsin[12,13] have provided several key insights into the conformational changes that occur on receptor activation. These structures confirm the large outward rotation of H6, the signature of an active receptor[14]. However, despite the large change in retinal configuration and orientation, the active-state crystal structures of rhodopsin show almost no change in structure on the extracellular side of the receptor when compared with the large changes observed on the intracellular side[15]. This observation is surprising as a substantial amount of absorbed light energy ($\sim 35 \, \mathrm{kcal \, mol}^{-1}$) is stored within retinal–protein interactions in the primary photoproduct bathorhodopsin[16] and then released as the retinal and surrounding protein relax during the transition to the active metarhodopsin II (Meta-II) intermediate[2]. The lack of structural changes in the binding site surrounding the retinal raises the question of how retinal isomerization and PSB deprotonation generate the large helix rearrangements on the intracellular side to create the intracellular G protein-binding pocket.

We take advantage of solid-state nuclear magnetic resonance (NMR) and Fourier transform infrared (FTIR) spectroscopy of Meta-II to address how protein residues within the 11-*cis* retinal-binding site adapt to the constrained all-*trans* retinal configuration following retinal isomerization. We also gain insight into how hydrogen-bonding networks on the extracellular surface of rhodopsin rearrange in response to PSB deprotonation. Low temperature (below $\sim 0 \, ^\circ \mathrm{C}$) slows the thermal steps in the rhodopsin photoreaction and provides a way to trap the native light-activated Meta-II state[17–19]. In contrast, high-resolution crystal structures, which capture elements of the active state, rely on soaking crystals of the apoprotein opsin with all *trans*-retinal[20] or have required stabilizing and/or activating mutations[12,13]. Crystals of native rhodopsin crack and no longer diffract following absorption of light and conversion to an active conformation[21]. Structural information from solid-state NMR experiments comes from distance measurements between specific $^{13}$C labels incorporated into the retinal and the protein, as well as from chemical shift measurements to assess changes in hydrogen bonding. Nevertheless, this approach relies on the crystal structures as a reference to establish how specific sites change position or orientation on activation. FTIR difference spectroscopy provides a complementary approach to NMR for assessing how specific residues in close association with the retinal control the equilibrium between Meta-I and Meta-II, the final step in the reaction pathway[22–24].

The first major observation to emerge from these studies is that the orientation of the retinal differs in Meta-II trapped along its reaction coordinate compared with Meta-II stabilized for crystallography. NMR constraints indicate that the β-ionone ring does not change orientation on activation, but rather the largest change in retinal structure involves rotation of the C20 methyl group toward extracellular loop-2 (EL2)[18]. In contrast, crystal structures obtained by diffusing all-*trans* retinal into opsin (Meta-II-opsin[20]) or using the constitutively active M257Y mutant[25] (Meta-II-M257Y; ref. 13) suggest that the all-*trans* retinal SB is in an orientation roughly opposite to that observed by NMR (Supplementary Fig. 1). The second major observation is that there are larger changes in the position of key aromatic residues (Tyr191[EL2] and Tyr268[6.51]) surrounding the retinal than observed in the crystal structures. These residues are highly conserved in the visual receptor subfamily. FTIR studies of the Y191F and Y268F mutants indicate that these tyrosines control the Meta-I ⇔ Meta-II equilibrium. We propose an activation mechanism that builds on these observations and previous studies by emphasizing the changes in extracellular residues in close proximity to the retinal. The mechanism that emerges from these studies implicates steric interactions as the dominant force driving structural changes in the light-triggered transition of rhodopsin to Meta-I, while electrostatic (and hydrogen bonding) interactions control the formation of the active Meta-II conformation.

## Results

**Retinal orientation in Meta-II**. To establish the orientation of the retinal in the binding site of the active Meta-II intermediate, we focus on the positions of the retinal C18, C19 and C20 methyl groups relative to surrounding protein residues (Fig. 1 and Supplementary Fig. 1). The C19 and C20 methyl groups extend from the retinal polyene chain, while the C18 methyl group extends from the β-ionone ring. In the crystal structures of Meta-II obtained using the M257Y mutant[13] or with opsin containing the bound retinal SB[20], the β-ionone ring has rotated relative to its position in rhodopsin, with the C18 methyl group facing the extracellular surface. In these models, the closest tyrosine residue to the C18 methyl group is Tyr191[EL2] (Supplementary Note 1 and Supplementary Tables 1–5). Below, we use NMR to measure a few specific distances between the retinal and surrounding protein to define the orientation of the retinal in Meta II. These constraints along with the results from previous biophysical and biochemical studies are used to suggest a general mechanism of receptor activation.

NMR distance measurements between $^{13}$C sites in the retinal and surrounding protein are made using dipolar-assisted rotational resonance (DARR) of rhodopsin and Meta-II (Supplementary Fig. 2). To address the orientation of the

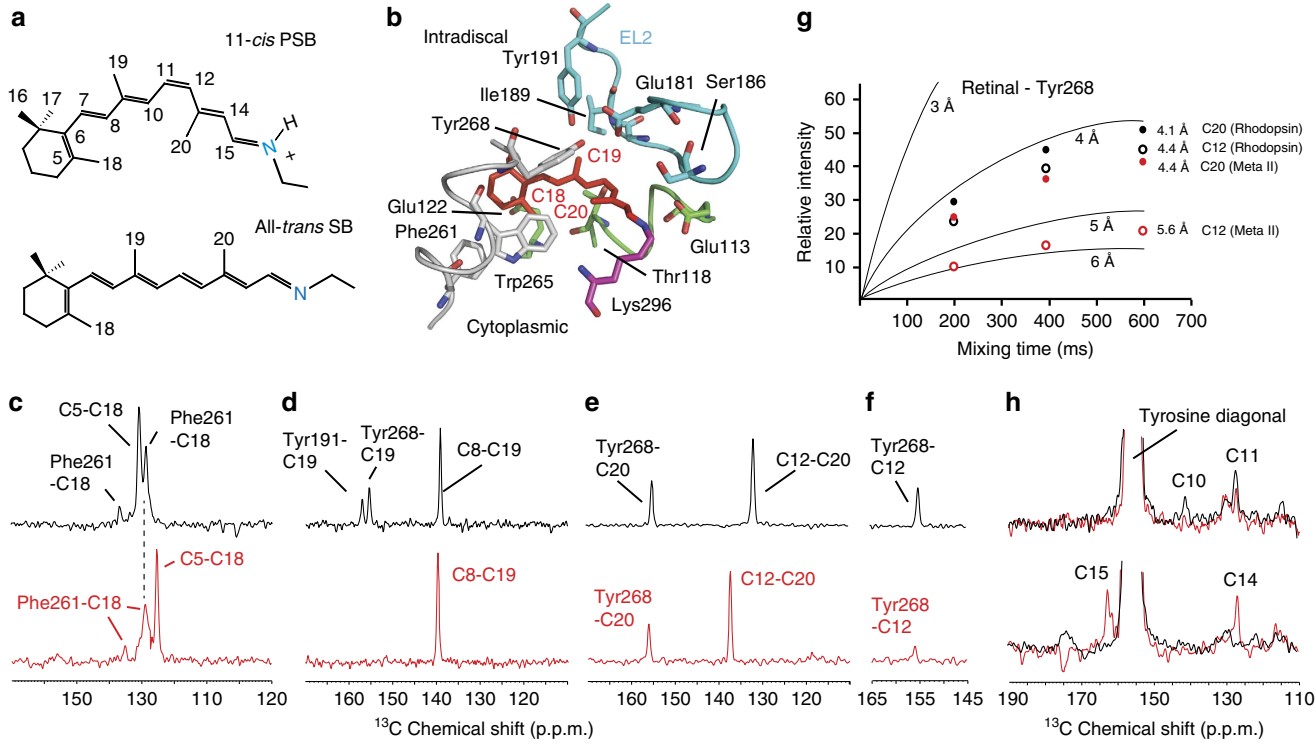

**Figure 1 | Retinal–protein contacts in dark-state rhodopsin and Meta-II. (a)** Structure of the 11-*cis* retinal PSB and all-*trans* retinal SB in rhodopsin and Meta-II, respectively. **(b)** Crystal structure of rhodopsin (PDB-ID 1U19; ref. 26) showing interactions of the C18, C19 and C20 retinal methyl groups with surrounding residues. **(c)** Retinal C18 contacts with phenylalanine and tyrosine in rhodopsin and Meta-II. Rows through the [13]C18 diagonal resonance are shown from [13]C DARR NMR experiments of rhodopsin (black) and Meta-II (red) obtained with the receptor regenerated with [13]C5, [13]C18 retinal and incorporating [13]C-ring-labelled phenylalanine and [13]Cζ-labelled tyrosine. Since the chemical shifts can change between rhodopsin and Meta-II, the rows selected correspond to the diagonal chemical shifts of the [13]C18 resonance in rhodopsin in 21.6 p.p.m. and Meta-II at 20.9 p.p.m. These rows maximize the crosspeak intensities. **(d)** Retinal C19 contacts with tyrosine. Rows through the [13]C19 diagonal resonances in rhodopsin (black) at 14.7 p.p.m. and Meta-II (red) at 13.8 p.p.m. obtained with the receptor regenerated with [13]C8, [13]C19 retinal and incorporating [13]C-ring Phe and [13]Cζ-labelled Tyr. **(e,f)** Retinal C12, C20 contacts with tyrosine in rhodopsin and Meta-II. Rows through the [13]C20 diagonal resonances in rhodopsin (black) at 16.4 p.p.m. and Meta-II (red) at 13.7 p.p.m. obtained with the receptor regenerated with [13]C12, [13]C20 retinal and incorporating [13]Cζ-labelled Tyr. **(g)** Build-up curves for the observed retinal–tyrosine crosspeaks in rhodopsin (black) and Meta-II (red) scaled to the C12-C20 crosspeaks. The C12-C20 retinal distance is fixed at ~2.4 Å. The retinal C12 and C20 distances to the Cζ carbon of Tyr268[6.51] range from ~4 to 6 Å (Supplementary Table 3). The retinal-Tyr268[6.51] data are compared with build-up curves derived from model compounds and fixed distances in rhodopsin. The error in the measurements is ± 0.3 Å on the basis of the signal-to-noise of the NMR spectra. **(h)** [13]Cζ-Tyr268[6.51] crosspeaks with the retinal [13]C10,11 resonances (top) and the [13]C14,15 resonances in rhodopsin (black) and Meta-II (red). Rows are taken through the [13]Cζ-Tyr diagonal resonance at 155.2 p.p.m. in rhodopsin and 156.1 p.p.m. in Meta-II.

retinal β-ionone ring, DARR NMR measurements were made using rhodopsin containing [13]C-ring-labelled phenylalanine and [13]Cζ-labelled tyrosine, and regenerated with [13]C5, [13]C18-labelled retinal. The diagonal [13]C resonances in two-dimensional (2D) DARR NMR spectra arise from [13]C-labelled amino acids and retinal. Crosspeaks appear between resonances when the distance between the [13]C atoms is <6–6.5 Å.

Rows taken from 2D NMR spectra of rhodopsin (Fig. 1c, top panel, black) and Meta-II (Fig. 1c, bottom panel, red) correspond to the regions containing crosspeaks between [13]C resonances. In rhodopsin, we observe an intense crosspeak between the two directly bonded retinal carbons, [13]C5 at 131.0 p.p.m. and [13]C18 at 21.6 p.p.m. There are also crosspeaks at ~129 and 137 p.p.m. that correspond to the chemical shifts of the aromatic ring carbons of [13]C-labelled phenylalanine. Phe261[6.44] is the only phenylalanine within the upper distance limit (~6.5 Å) of the DARR measurements (Supplementary Tables 1–5). The intensities of the crosspeaks are consistent with the short internuclear distances (4.9–5.4 Å) between the retinal C18 methyl and the ring carbons of Phe261[6.44] observed in the crystal structure of rhodopsin (PDB-ID 1U19)[26].

On conversion to Meta-II, the retinal [13]C resonances change frequency due to changes in retinal structure, protonation state and retinal–protein interactions. For example, the crosspeak between the [13]C5 and [13]C18 resonances shifts from 131 to 126 p.p.m. (Fig. 1c), but the crosspeak intensity remains the same because of the fixed distance between these two carbons. The intra-retinal C5-C18 crosspeak provides an internal control for calibrating the intensity changes observed in protein–retinal crosspeaks. The intra-retinal [13]C8-[13]C19 and [13]C12-[13]C20 crosspeaks provided similar internal controls in Fig. 1d,e. The intensity of the crosspeaks between the retinal and Phe261[6.44] ring-[13]C resonances remains approximately the same as in rhodopsin, indicating that there is little change in the distance between Phe261[6.44] and the [13]C18 methyl group. This observation argues that the β-ionone ring does not flip as suggested by the Meta-II crystal structures. The position of the β-ionone ring is additionally constrained by contacts between the [13]C5, [13]C18 and the [13]C16, [13]C17 retinal resonances and residues (Met207[5.42] and His211[5.46]) on H5 (Supplementary Fig. 3 and Supplementary Note 2). Furthermore, in the Meta-II-opsin and Meta-II-M257Y crystal structures, the rotation of the β-ionone ring is predicted to bring the C18 methyl group to within the DARR distance limit

($\sim$6 Å) of $^{13}$Cζ-Tyr191$^{EL2}$, which is not observed in the present studies (Fig. 1c).

The C19 methyl group provides another probe of the retinal orientation. In rhodopsin, the C19 methyl is oriented towards EL2 and packed between Thr118$^{3.33}$, Tyr191$^{EL2}$, Ile189$^{EL2}$ and Tyr268$^{6.51}$ (Fig. 1b). We observe two C19-$^{13}$Cζ-tyrosine crosspeaks at 155.2 and 156.6 p.p.m. (C19-Tyr268$^{6.51}$, 4.3 Å; and C19-Tyr191$^{EL2}$, 4.7 Å) (Fig. 1d). In Meta-II, the $^{13}$C19-$^{13}$Cζ-tyrosine crosspeak intensity is lost, consistent with an increased separation between these tyrosines and the retinal C19 methyl group to $> \sim$6–6.5 Å. We had previously attributed the loss of C19-tyrosine crosspeak intensity to translational motion of the retinal towards H5 and motion of EL2 (ref. 27). Below, we now show that the chemical shifts of both Tyr191$^{EL2}$ and Tyr268$^{6.51}$ change in Meta-II, suggesting that they move as well. This contributes to the loss of crosspeak intensity between the retinal C19 methyl resonance and the overlapping $^{13}$Cζ-Tyr191$^{EL2}$ and Tyr268$^{6.51}$ resonances. Moreover, we show that the alternative explanation, a flip of the β-ionone ring, does not occur until Meta-II decays (Supplementary Fig. 4 and Supplementary Note 3).

The retinal C20 methyl group is twisted out of the plane of the retinal polyene chain and primed to rotate in the clockwise direction when viewed from the Schiff's base end of the retinal[28–30]. We previously concluded that the C20 methyl rotates towards EL2 on the basis of a C20-Tyr268$^{6.51}$ crosspeak[18,19]. Here NMR measurements on the orientation of the C20 methyl group were undertaken with rhodopsin $^{13}$C-labelled at the ζ-carbon of tyrosine and regenerated with $^{13}$C12, $^{13}$C20 retinal (Fig. 1e,f). The row from the 2D DARR NMR spectrum of rhodopsin (black) through the C20 diagonal resonance yields an intense C12-C20 crosspeak at 132.2 p.p.m. (internal control) and a strong crosspeak to $^{13}$Cζ-Tyr268$^{6.51}$ at 155.2 p.p.m. (refs 18,19). On conversion to Meta-II (red), the C20-Tyr268$^{6.51}$ crosspeak is only modestly weaker relative to the C12-C20 intra-retinal crosspeak, whereas the C12-Tyr268$^{6.51}$ crosspeak becomes markedly weaker.

The crosspeak intensities between the $^{13}$Cζ-Tyr268$^{6.51}$ and the retinal $^{13}$C12,$^{13}$C20 resonances constrain the position of the tyrosine side chain relative to the retinal. The large difference in intensity in Meta-II between the C12 and C20 crosspeaks with Tyr268$^{6.51}$ indicate that the C20 methyl group is closer to the tyrosine-Cζ carbon than the C12 methyl group. Molecular dynamics simulations performed on the basis of previous NMR constraints[28] generated similar distances ($\sim$4.3 Å) between Tyr268$^{6.51}$ and the C12 and C20 carbons in Meta-II (Supplementary Table 3). The current NMR measurements indicate that the C20-Tyr268$^{6.51}$ separation ($\sim$4.4 Å) is shorter than the C12-Tyr268$^{6.51}$ separation ($\sim$5.6 Å). In contrast, the X-ray structures of Meta-II show the reverse, namely, the C12-Tyr268$^{6.51}$ separation is shorter ($\sim$3.9 Å) than the C20-Tyr268$^{6.51}$ separation ($\sim$5.9 Å). To accommodate the NMR distances, we propose (see below, Supplementary Fig. 5 and Supplementary Note 4) that H6 tilts inward on activation, a motion that is not captured in the molecular dynamics simulations.

Figure 1g presents build-up curves of crosspeak intensity for the retinal C12 and C20 carbons and Tyr268$^{6.51}$. These curves provide a more accurate measure of internuclear distance than measurements at a single mixing time[31,32]. The distances in rhodopsin (black symbols) derived from this experiment are consistent with crystal structure distances (C12-Tyr268$^{6.51}$, 4.5 Å; C20-Tyr268$^{6.51}$, 4.1 Å; PDB-ID 1GZM (ref. 33)). In contrast, in Meta-II (red symbols) there is a marked difference between the distances estimated on the basis of the NMR build-up curves and the distances taken from the Meta-II crystal structures

(Supplementary Table 3). For example, in the Meta-II-opsin and Meta-II-M257Y structures, the C12 carbon is much closer to the extracellular surface and consequently is in closer proximity to Tyr268$^{6.51}$ than C20. The crystal structures place C12 at a distance of 3.8–4.0 Å from Tyr268$^{6.51}$ (Supplementary Table 3), which should give rise to a stronger NMR crosspeak than observed. The intensity of the C12-Tyr268$^{6.51}$ crosspeak drops considerably between rhodopsin and Meta-II indicating an increase in distance of C12 from Tyr268$^{6.51}$. The NMR-derived distance is estimated to be $\sim$5.6 ± 0.3 Å. Together, the NMR distance constraints presented above indicate that the orientation of the retinal in Meta-II trapped along its reaction coordinate is different than in the crystal structures.

**Tyr268 movement suggests an inward tilt of H6 in Meta-II.** Tyr268$^{6.51}$ has the highest subfamily conservation (97%) in the visual GPCRs after Lys296$^{7.43}$, the site of retinal attachment, indicating that its position and interactions are critically important within the visual receptors. A change in the position of Tyr268$^{6.51}$ in Meta-II is detected by monitoring tyrosine crosspeaks to the retinal $^{13}$C10, $^{13}$C11 and the $^{13}$C14, $^{13}$C15 carbons. Tyr268$^{6.51}$ exhibits crosspeaks to both the retinal $^{13}$C10 and $^{13}$C11 resonances in rhodopsin (Fig. 1h). These carbons are $\sim$4 Å from the Cζ position of Tyr268$^{6.51}$ in the rhodopsin crystal structures. In contrast, no crosspeak intensity is observed between the retinal $^{13}$C14, $^{13}$C15 resonances and the $^{13}$Cζ resonance of Tyr268$^{6.51}$ in rhodopsin. On conversion to Meta-II, the intensities of both the $^{13}$C10 and $^{13}$C11 crosspeaks to Tyr268$^{6.51}$ decrease, while the $^{13}$C14 and $^{13}$C15 crosspeaks increase. Together, these changes indicate a shift in the position of Tyr268$^{6.51}$ towards the Schiff base linkage in Meta-II.

The shift of Tyr268$^{6.51}$ towards the retinal C14 and C15 carbons argues for a location between the retinal and EL2. We also observe strong chemical shift changes in the region of the SB consistent with a change in its electrostatic environment[34]. The retinal C15 = N bond is strongly polarized; a large downfield chemical shift of C15 on conversion to Meta-II indicates that this carbon bears a substantial partial positive charge. The C = N polarization suggests close interactions with Glu113$^{3.28}$, Glu181$^{EL2}$ and/or Tyr268$^{6.51}$.

The movement of Tyr268$^{6.51}$ towards EL2 is supported by changes in the intensity of crosspeaks to Gly188$^{EL2}$ and Cys187$^{EL2}$ in Meta-II (Fig. 2). We have previously shown that Tyr268$^{6.51}$ exhibits a weak crosspeak with Gly188$^{EL2}$ in rhodopsin[27]. 2D NMR spectra using rhodopsin containing $^{13}$Cα-Gly, $^{13}$Cζ-Tyr and $^{13}$C = O-Cys allow us to highlight the Cys187$^{EL2}$-Gly188$^{EL2}$ pair since the Cys-$^{13}$C = O to Gly-$^{13}$Cα distance is short (2.4 Å) and consequently yields an intense crosspeak. As with the double $^{13}$C-labelled retinals, the Cys187$^{EL2}$-Gly188$^{EL2}$ pair serves as an internal control since the distance between these labelled $^{13}$C sites does not change between rhodopsin and Meta-II (Fig. 2b). On conversion to Meta-II, we observe two $^{13}$Cα-Gly-$^{13}$Cζ-Tyr crosspeaks (Fig. 2b). The weaker crosspeak at 44.5 p.p.m. in Meta-II is assigned to Gly114$^{3.29}$-Tyr178$^{EL2}$ on the basis of the Y178F mutant (Fig. 2f). The stronger crosspeak at 43.1 p.p.m. is assigned to an overlap of the Gly188$^{EL2}$-Tyr268$^{6.51}$ crosspeak and at least one other Tyr-Gly contact. There is considerable loss of intensity at this position in the G188A mutant, and complete loss of the Cys187$^{EL2}$-Gly188$^{EL2}$ crosspeak (Fig. 2e).

The Cys187$^{EL2}$ $^{13}$C = O label provides an additional probe for the location of Tyr268$^{6.51}$ in Meta-II. There are two Cys-Tyr pairs in rhodopsin that are <6 Å apart (Tyr136$^{3.51}$-Cys140$^{3.55}$, 5.1 Å; and Tyr206$^{5.41}$-Cys167$^{4.56}$, 5.9 Å; PDB-ID 1U19 (ref. 26)). Both Tyr-Cys distances increase to >6 Å in the opsin crystal

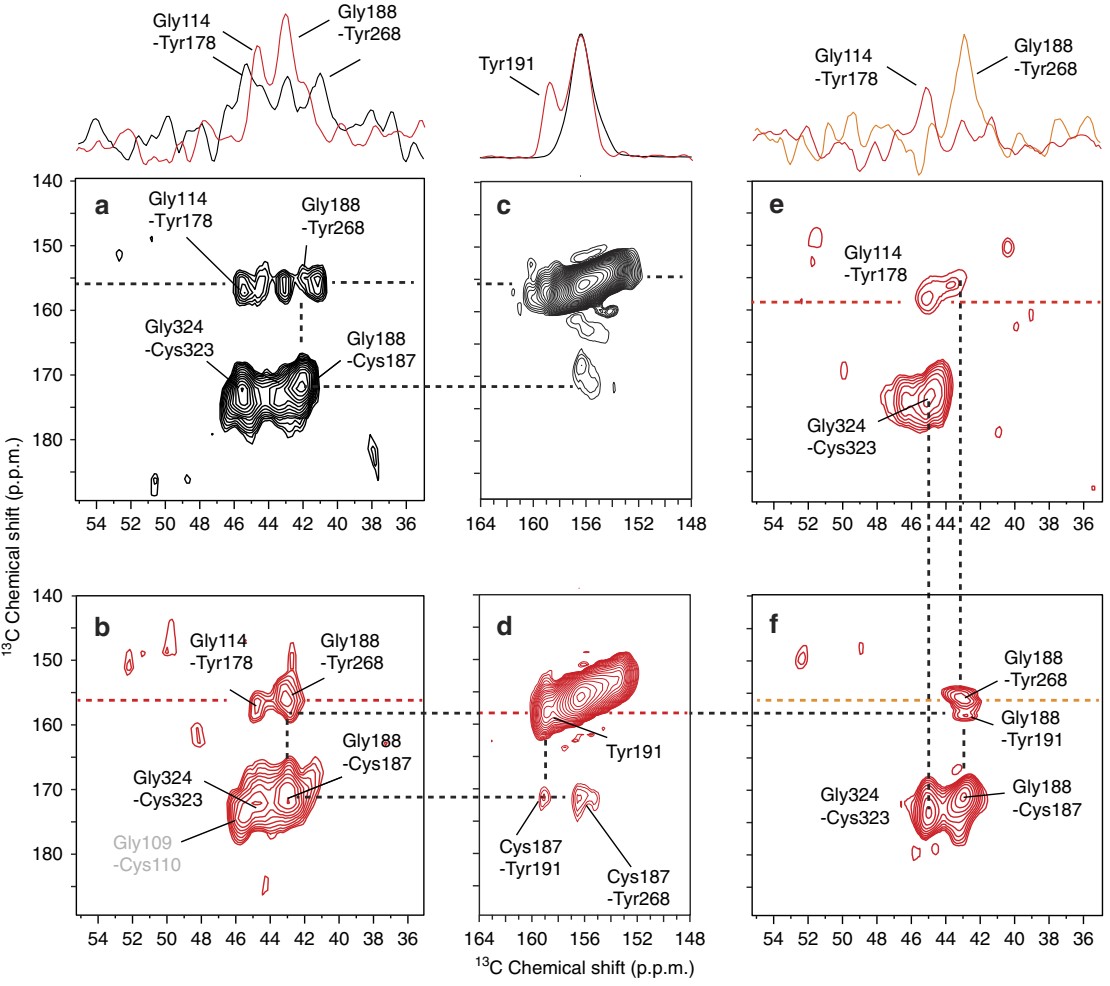

**Figure 2 | Tyr268$^{6.51}$ to Gly188$^{EL2}$ and Tyr268$^{6.51}$ to Cys187$^{EL2}$ contacts in Meta-II.** (**a**) 2D DARR spectrum highlighting Tyr-Gly contacts in rhodopsin using rhodopsin containing $^{13}$Cζ-Tyr, $^{13}$Cα-Gly and $^{13}$C=O-Cys. In the rhodopsin crystal structure (PDB-ID 1U19; ref. 26), there are six Tyr(Cζ)-Gly(Cα) contacts, which are all located in the extracellular region of rhodopsin. These contacts involve five tyrosines and five glycines: Tyr10$^{N-term}$-Gly3$^{N-term}$, 3.9 Å; Tyr10$^{N-term}$-Gly280$^{EL3}$, 4.4 Å; Tyr29$^{N-term}$-Gly101$^{EL1}$, 4.0 Å; Tyr178$^{EL2}$-Gly114$^{3.29}$, 4.5 Å; Tyr191$^{EL2}$-Gly188$^{EL2}$, 5.2 Å; and Tyr268$^{6.51}$-Gly188$^{EL2}$, 5.3 Å. Above **a** are shown rows through the Tyr-Gly crosspeaks. The rows better illustrate the intensity change occurring in the Tyr268$^{6.51}$-Gly188$^{EL2}$ peak on activation. The observation that the Tyr178$^{EL2}$-Gly114$^{3.29}$ crosspeak does not change intensity is consistent with the lack of influence of the Y178F mutation on the Meta-I–Meta-II transition (Fig. 4g). (**b**) Wild-type Meta-II spectrum using rhodopsin containing $^{13}$Cζ-Tyr, $^{13}$Cα-Gly and $^{13}$C=O-Cys. (**c,d**) Region of the 2D DARR spectra of rhodopsin (black) and Meta-II (red) corresponding to $^{13}$Cζ-Tyr to $^{13}$C=O-Cys crosspeaks. Above **c** are shown rows through the $^{13}$Cζ-Tyr diagonal of rhodopsin and Meta-II. (**e,f**) G188A Meta-II and Y178F Meta-II spectra, respectively, using the same $^{13}$C-labelling scheme as above. Rows are shown through the Tyr-Gly crosspeaks in the G188A Meta-II and Y178F Meta-II spectra above **e**.

structure[6]. We observe a weak crosspeak in rhodopsin between a Cys-$^{13}$C=O resonance (168–172 p.p.m.) and a Tyr-$^{13}$Cζ resonance (~156 p.p.m.; Fig. 2c). We observe two stronger crosspeaks in Meta-II at 172.5 p.p.m. (Fig. 2d), the same shift as the Cys187$^{EL2}$ $^{13}$C=O defined in Fig. 2b. One of the tyrosine contacts can be assigned to Tyr191$^{EL2}$ on the basis of its unique $^{13}$Cζ chemical shift at 159.3 p.p.m. (ref. 18). We assign the other crosspeak to a Cys187$^{EL2}$-Tyr268$^{6.51}$ interaction.

The position of the $^{13}$Cζ-Tyr268$^{6.51}$ relative to $^{13}$C15-labelled retinal, as well as to $^{13}$Cα-Gly188 and 1-$^{13}$C-Cys187, indicates that Tyr268$^{6.51}$ has shifted in the retinal-binding site. The separation between the $^{13}$Cζ Tyr268$^{6.51}$ carbon and these sites is >6 Å in the Meta-II crystal structures, but much shorter in Meta-II trapped in the NMR experiments. The shorter distance is consistent with a small inward tilt of the extracellular end of H6. In addition to the crosspeak of retinal C20 with Tyr268$^{6.51}$ in Meta-II (Fig. 1e), the C20 methyl group also exhibits a crosspeak with a glycine having a chemical shift of 46.5 p.p.m., which is assigned to Gly121$^{3.36}$ on the basis of mutational studies

(Supplementary Fig. 5 and Supplementary Note 4). Together, the retinal C20-Tyr268$^{6.51}$ and -Gly121$^{3.36}$ contacts indicate that the orientation of the C20 methyl group is roughly halfway between the orientation found in the Meta-II crystal structures and the orientation that previously resulted from molecular dynamics simulations (Supplementary Fig. 1).

**Extracellular tyrosine switch releases constraints on H6.** Tyr268$^{6.51}$ on TM helix H6 is strongly hydrogen-bonded to Glu181$^{EL2}$ on the β3 strand of EL2 and to Tyr191$^{EL2}$ on the β4 strand of EL2 (Fig. 3d and Supplementary Table 5). EL2 caps the extracellular side of the retinal-binding site and Glu181$^{EL2}$ has previously been identified as a second counter-ion to the retinal PSB, shifting position in the transition of rhodopsin to the Meta-I intermediate[35]. As a result, Tyr268$^{6.51}$ is in a key location to couple hydrogen-bonding changes involving EL2 and the retinal PSB with motion of H6. The NMR data discussed above show that Tyr268$^{6.51}$ has shifted relative to the retinal in Meta-II, while

the large downfield chemical shift of Tyr191[EL2] suggests that it has shifted relative to Glu181[EL2].

To confirm that the changes in Tyr191[EL2] and Tyr268[6.51] are triggered by the shift in the position of Glu181[EL2], we obtained both one-dimensional (1D) difference NMR spectra and 2D DARR NMR spectra (Supplementary Fig. 6 and Supplementary

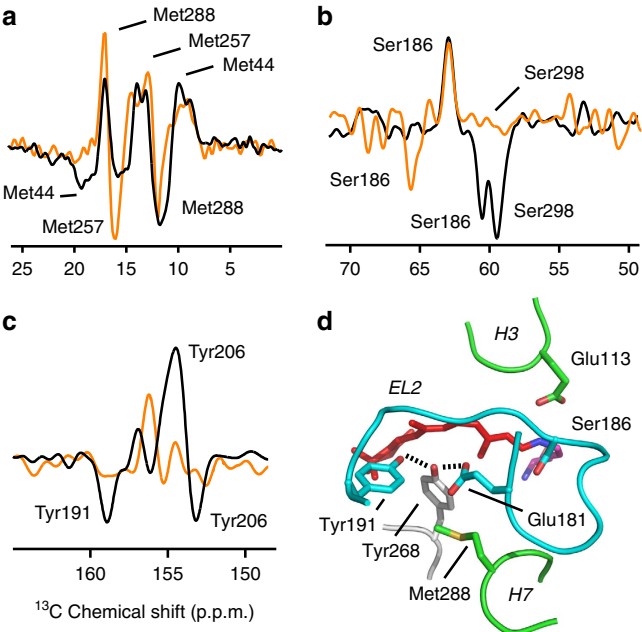

**Figure 3 | Changes in EL2 interactions in Meta-I and Meta-II.** 1D NMR difference spectra are shown of rhodopsin minus Meta-I (orange) and rhodopsin minus Meta-II (black) using rhodopsin containing labelled $^{13}$Cε-Met (**a**), $^{13}$Cβ-Ser (**b**) or $^{13}$Cζ-Tyr (**c**). The rhodopsin spectra correspond to positive peaks, while the Meta-I and Meta-II spectra correspond to negative peaks. The assignments of the resonances have previously been reported[18,19,38,64]. (**d**) Crystal structure of rhodopsin in the region of Tyr268[6.51] (PDB-ID 1U19; ref. 26).

Note 5) focusing on residues in close proximity to Glu181[EL2]. 1D NMR difference spectra of $^{13}$Cε-Met-labelled rhodopsin show that the side-chain methyl group of Met288[7.35] has an unusual downfield chemical shift in rhodopsin, which moves upfield in both Meta-I and Meta-II (Fig. 3a). Met288[7.35] represents a contact point between H7 and EL2, and is in close proximity to Glu181[EL2] in the rhodopsin structure. These chemical shift changes are consistent with the shift of the negatively charged Glu181[EL2] side chain away from the methyl group of Met288[7.35]. Motion of Glu181[EL2] towards the retinal PSB suggests that it comes into close proximity with Ser186[EL2]. 1D difference spectra of $^{13}$Cβ-labelled-serine reveal a downfield shift in Meta-I and upfield shift in Meta-II (Fig. 3b). The chemical shift of the $^{13}$Cβ carbon of serine is sensitive to Cβ-OH hydrogen bonding, and the observed chemical shift changes are consistent with increased hydrogen bonding in Meta-I (or earlier photointermediate)[36] and decreased hydrogen bonding in Meta-II.

In rhodopsin, the chemical shifts of both $^{13}$Cζ-Tyr191[EL2] and $^{13}$Cζ-Tyr268[6.51] are at ~156 p.p.m. (Fig. 1d). There is a positive peak in the rhodopsin–Meta-I difference spectrum at this position, but no corresponding negative peaks (Fig. 3c). In contrast, we have previously assigned[18,19] the downfield resonance at 159.3 p.p.m. in the rhodopsin–Meta-II difference spectrum to Tyr191[EL2] and have suggested that this unusual chemical shift is due to direct interaction with Glu181[EL2].

Together, the chemical shift changes of Met288[7.35] and Ser186[EL2] indicate that the counter-ion shift of Glu181[EL2] in Meta-I precedes the motion of Tyr191[EL2] in Meta-II. Tyr191[EL2] is tightly packed against Ala272[6.55] on H6, one helical turn from Tyr268[6.51], and its motion towards Glu181[EL2] would release steric constraints hindering the inward tilt of the extracellular end of H6 in the dark state.

**Conserved extracellular tyrosine residues stabilize Meta-I.** The interactions involving Tyr191[EL2], Tyr268[6.51] and Met288[7.35] indicate that these residues mediate the transition between Meta-I and Meta-II. To further explore their influence on activation, we undertook FTIR studies on the Y191F, Y268F and M288L mutants alongside wild-type rhodopsin. We previously showed

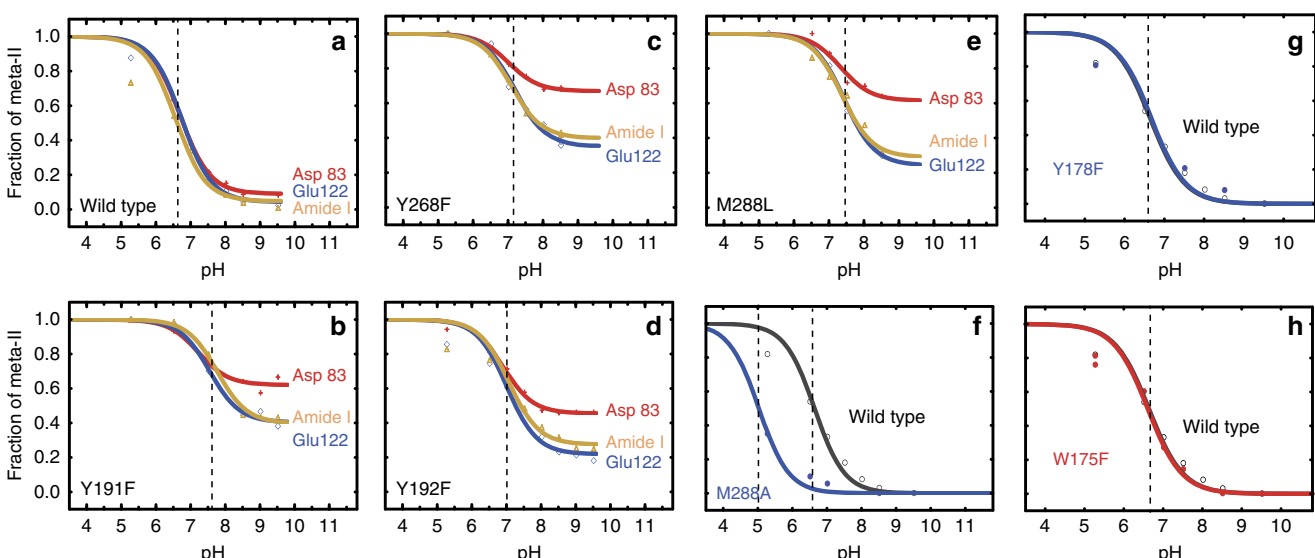

**Figure 4 | FTIR analysis of rhodopsin.** Wild-type (**a**), and the Y268F (**b**), Y191F (**c**), Y192F (**d**), M288L (**e**), M288A (**f**), Y178F (**g**) and W175F (**h**) mutants. FTIR difference spectra were obtained at 0 °C as a function of pH, and the relative contributions of Meta-I and Meta-IIbH+ were determined on the basis of reference spectra (Supplementary Fig. 7). Spectral decomposition was performed between 1,800 and 1,600 cm$^{-1}$. This range is the most diagnostic of the Meta-I–Meta-II transition and comprises the amide I vibrations of the protein backbone and the C=O stretch of protonated carboxylic acids Glu122[3.37] and Asp83[2.50]. The vertical dashed lines are at the positions of the inflection points.

that the amide I infrared band along with the $C=O$ stretching modes of the protonated Glu122[3.37] and Asp83[2.50] carboxylic groups provide conformationally sensitive markers and can be used to deconvolute the Meta-I $\Leftrightarrow$ Meta-II equilibrium into the inactive Meta-I state and several Meta-II substates (Meta-IIa, Meta-IIb and Meta-IIbH+; Supplementary Fig. 7 and Supplementary Note 6)[22].

A two-state equilibrium is observed at 0 °C between Meta-I and Meta-IIbH+ using wild-type rhodopsin reconstituted into egg phosphocholine membranes, where the transition between Meta-I and Meta-IIbH+ has an apparent pK of ~6.8 and is similar for each of these FTIR markers (Fig. 4a). At higher temperature, the titration curves become more complex with stabilization of the Meta-IIb substate (Supplementary Fig. 7). The Y268F, Y191F and M288L mutants exhibit complex titration curves (Fig. 4b,c,e) even at 0 °C due to a strong forward shift of the Meta-I $\Leftrightarrow$ Meta-IIb equilibrium to Meta-IIb. For wild-type rhodopsin, an increase in temperature will lead to a shift of the equilibrium as the stabilization of Meta-IIb has an entropic component[22]. In contrast for the mutants, the forward shift reflects the loss of stabilizing interactions in Meta-I.

The Y192F mutant (Fig. 4d) exhibits a forward shift of the Meta-I $\Leftrightarrow$ Meta-IIb equilibrium as in Y191F, but the effect is not as large. The M288A mutant (Fig. 4f) reveals a downshift of the apparent pKa, indicating stabilization of the Meta-I intermediate on mutation. Thus, depending on the specific mutation at Met288[7.35], the Meta-I $\Leftrightarrow$ Meta-II equilibrium can be shifted in either direction. Finally, the equilibrium is not affected by the W175F and Y178F mutants (Fig. 4g,h) indicating that there are conserved aromatic residues in EL2 that do not have the effects observed for Tyr191[EL2] and Tyr268[6.51].

## Discussion

Activation of the visual receptors is triggered by changes in both steric and electrostatic interactions caused by 11-*cis* to all-*trans* isomerization of the retinal and deprotonation of the retinal PSB. Retinal isomerization occurs in the first step of the rhodopsin photoreaction, while deprotonation occurs on formation of the Meta-II intermediate[37]. The specific retinal–protein interactions observed by NMR and FTIR provide insights into how these two components of the activation mechanism work together to drive the outward tilt of TM helix H6. We liken these connected activation events to the firing mechanism of a two-stage trigger.

The first stage of the trigger results from isomerization-induced steric contact between the β-ionone ring and residues on helices H3, H5 and H6 (Fig. 5). NMR[38] and fluorescence[39] studies both reveal that the first major change in receptor conformation is rotation of H6 in the formation of Meta-I, which breaks the intracellular Arg135[3.50]-Glu247[6.30] ionic lock[40] and results in Trp265[6.48] displacement[38,41]. These rearrangements are accompanied by small changes at the cytoplasmic end of H5 (refs 39,42). Our current results indicate direct contact of the β-ionone ring with His211[5.46] and Phe261[6.44] in Meta-II (Fig. 5a). Previously, we reported that activation results in a loss of the retinal C18-Trp265[6.48] contact[19]. Mutation of either Phe261[6.44] or Trp265[6.48] lowers the initial rate of G protein transducin activation, ~20–60% for F261A rhodopsin[43] and ~90% in W265F rhodopsin[44], consistent with their roles in the activation mechanism. The steric contact of the β-ionone ring with Phe261[6.44] observed in Meta-II suggests that the phenylalanine ring acts a lever for rotation of H6. This residue is part of a transmission switch in the conserved TM core of GPCRs[15,45]. The conserved proline (Pro215[5.50]) on H5 is also a component of the transmission switch and results in a free backbone carbonyl at His211[5.46]. This carbonyl forms an interhelical hydrogen bond

with Glu122[3.37] in rhodopsin, which is replaced in Meta-II by a direct interaction between the His211[5.46] and Glu122[3.37] side chains[20]. The strong steric contacts of the β-ionone ring with His211[5.46] and Glu122[3.37] that drive this transition allow H5 to reorient in Meta-II (ref. 46). Mutation of Glu122[3.37] breaks their direct interaction and destabilizes Meta-II (ref. 47). Retinal analogues in which the ring is truncated shift the conformational equilibrium between the Meta-I and Meta-II intermediates towards the inactive Meta-I state[24,48]. Thus, the β-ionone ring is a key element in both the first stage of the activation trigger and in stabilizing the Meta-II intermediate.

In Supplementary Fig. 4, we follow the change in orientation of the C18 methyl group of the β-ionone ring during the decay of Meta-II and observe a gain in contact with Tyr191[EL2] consistent with a flip of the β-ionone ring and suggesting that the crystal structures are capturing a relaxed, low-energy conformation that has retained the capacity to bind all-*trans* retinal. The final conformation observed by NMR is possibly Meta-III, in which the SB is reprotonated[49]. The observation of an all-*trans* retinal PSB following Meta-II suggests that the reprotonation event is the driving force for rotation along the long axis of the retinal during Meta-II decay. Together, our NMR results are consistent with the low temperature trapped Meta-II intermediate being a transient, high-energy state that is formed along the reaction coordinate and consequently captures the conformation triggering receptor activation.

The second stage in the two-stage trigger mechanism involves deprotonation of the retinal PSB along with steric interactions of the retinal C19 and C20 methyl groups with Tyr191[EL2] and Tyr268[6.51], respectively. Tyr191[EL2] and Tyr268[6.51] are part of an extensive hydrogen-bonding network that stretches through Glu181[EL2] to Glu113[3.28], the PSB counter-ion (Fig. 5b, Supplementary Fig. 8 and Supplementary Note 7). Isomerization results in motion of the retinal PSB proton away from the stabilizing interaction with Glu113[3.28]. This ultimately leads to proton transfer from the PSB to Glu113[3.28] in Meta-II. Nevertheless, the change in the position of the PSB proton must already be sensed by nearby residues in the Meta-I intermediate before proton transfer. The counter-ion shift mechanism envisions motion of the charged Glu181[EL2] side chain towards the PSB end of the retinal in Meta-I (ref. 35). The large changes in the NMR chemical shifts of Ser186[EL2] and Met288[7.35] observed in Meta-I are consistent with Glu181[EL2] motion. However, motion of Glu181[EL2] in Meta-I precedes the change in the position of Tyr191[EL2] and Tyr268[6.51] (Fig. 5b) because the striking downfield chemical shift of Tyr191[EL2] is not observed until Meta-II. Moreover, the proposed mechanism argues that the tilt of H6 occurs following PSB deprotonation and movement of Tyr191[EL2], in agreement with EPR measurements[50]. These results are consistent with an earlier FTIR study suggesting a role for Tyr268 and H6 reorientation in rhodopsin activation[51].

The C19 and C20 methyl groups contribute to the rearrangement of the extracellular hydrogen-bonding network. These methyl groups extend from the polyene chain and are located on opposite sides of the *cis* C11=C12 bond (Fig. 1a). The crystal structures of rhodopsin show that the C20 methyl group is primed to rotate in a clockwise direction on isomerization[28–30], the only potential obstacle being Tyr268[6.51]. The C20-Tyr268[6.51] crosspeak observed in Meta-II by NMR is consistent with this trajectory of the C20 methyl group and suggests that a steric contact with Tyr268[6.51] contributes to the rearrangement of the hydrogen-bonding network stabilizing Meta-I on the extracellular surface of the receptor. In rhodopsin regenerated with an 11-*cis* retinal analogue lacking the C20 methyl group, the photoreaction is slowed[52] and the quantum yield is reduced[53].

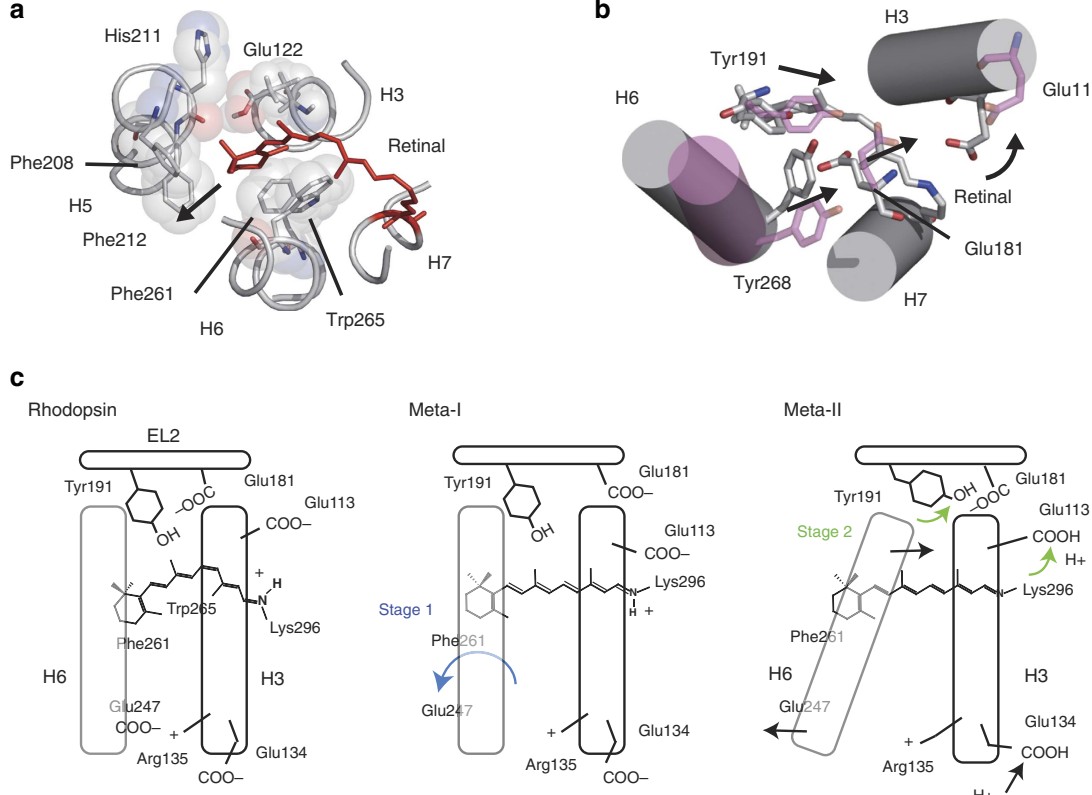

**Figure 5 | Two-stage trigger mechanism for rhodopsin activation. (a)** Residues interacting with the β-ionone ring in the dark state of rhodopsin (PDB-ID 1U19; ref. 26). The first stage of the two-stage trigger involves isomerization of the 11-*cis*-retinal chromophore within a tightly packed retinal-binding pocket. Steric interactions arise because the shape of the retinal-binding site fits the 11-*cis* isomer but does not accommodate the longer all-*trans* form[10]. The position of the β-ionone ring defined by distance measurements with Met207[5.42] and His211[5.46] on H5 (Supplementary Fig. 3 and refs 18,19), as well as Phe261[6.44] and Trp265[6.48] on H6 (Fig. 1 and refs 18,19,63) are consistent with strong steric interactions between the β-ionone ring in this region of the retinal-binding site. Motion of H5 and H6 is present at the Meta I stage as a result of isomerization[38,39,42]. **(b)** Cartoon representation of the structural changes occurring on the extracellular side of the retinal-binding site in rhodopsin comprising the second stage of the two-stage trigger. Retinal isomerization leads to a change in the position of Glu181[EL2] in Meta-I (ref. 35). Deprotonation of the retinal PSB and protonation of Glu113[3.28] leads to a further rearrangement of the hydrogen-bonding network connecting Glu113[3.28] to Tyr191[EL2]. Both Tyr268[6.51] and Tyr191[EL2] shift towards Glu181[EL2] to maintain hydrogen-bonding interactions. The proposed motion of the extracellular side of H6 due to Tyr191[EL2] motion is coupled to the outward pivot of the intracellular side of H6. **(c)** Schematic representation of the two-stage trigger mechanism.

The C19 methyl group of retinal is oriented towards EL2 in the dark state of rhodopsin and is tightly packed between Ile189[EL2] and Tyr191[EL2] (Supplementary Fig. 8). Retinal analogues lacking the C19 methyl group cannot activate rhodopsin[54], while increasing the size of the C19 methyl to either an ethyl or propyl group increases receptor activity in the dark, even though the retinal still remains covalently bound and in its 11-*cis* configuration[43,55,56]. On isomerization, the C19 methyl group rotates in the opposite direction (that is, anticlockwise) to the C20 methyl group[28–30], and consequently would have increased steric contact with Tyr191[EL2]. Removal of the C19 methyl group eliminates this interaction, while the ethyl or propyl substitutions at C19 result in a steric clash with Tyr191[EL2], which would disrupt the Tyr191[EL2]-Tyr268[6.51] hydrogen-bonding network in the dark, in a manner similar to light-driven isomerization in the wild-type receptor.

The net effect of PSB deprotonation and rearrangement of the hydrogen-bonding network involving EL2 is a shift of Tyr191[EL2] away from H6. We propose that this motion allows the extracellular end of H6 to pivot inward. In the visual pigments, Tyr191[EL2] has a high level of sequence identity (61%), and an overall level of conservation of 83% as either tyrosine or tryptophan. Interestingly, even though Tyr191[EL2] and Tyr268[6.51] contribute to Meta-I stability (Fig. 4), mutation of

either of these residues results in a substantial drop in G protein transducin activation[44,57]. This dual influence of Tyr191[EL2] and Tyr268[6.51] on Meta-I stability and Meta-II activity is consistent with one set of hydrogen-bonding interactions stabilizing Meta-I and a second set stabilizing Meta-II (Fig. 5). The observation that Tyr191[EL2] and Tyr268[6.51] strongly stabilize the Meta-I state suggests that in crystal structures of opsin and/or Meta-II, the inactive-state hydrogen-bonding network reforms in this region under the conditions used for crystallization.

The proposed two-stage trigger mechanism builds on the global toggle switch mechanism developed on the basis of ligand-activated GPCRs, which envisions a vertical see-saw movement of helix H6 with conserved Pro267[6.50] serving as the pivot point[58]. In the case of the visual pigments, Meta-I corresponds to the inactive state of a ligand-activated GPCR, where ligand binding simply allows H6 to pivot to open up the intracellular G protein-binding pocket.

## Methods

**Expression and purification of [13]C-labelled bovine rhodopsin.** Isotope-enriched bovine opsin[59] was expressed using inducible HEK293S cell lines[60], which are widely used for the production of recombinant proteins and viruses. The primers used for construction of the rhodopsin mutants are listed in Supplementary Table 6. The original cell lines were obtained from Jeremy Nathans (Johns Hopkins

University), but not authenticated or tested for mycoplasma contamination as they were used only for protein expression. The expressed opsin was generated into rhodopsin through incubation with ~30 μM 11-*cis* retinal and extracted from membranes using 1% (w/v) *n*-β-D-dodecyl maltopyranoside (DDM) in PBS pH 7.4. The detergent-solubilized lysates were separated from the remaining cell debris via centrifugation at 25,000*g* for 30 min at 4 °C. The detergent-solubilized lysate was applied to a rho-1D4-Sepharose column via gravity flow to capture rhodopsin. The flow-through volume was saved and the amount of unbound receptor assayed by western blot using rho-1D4. The column was then washed with 10 volumes of 0.02% DDM in PBS (pH 7.2). A subsequent wash step with 0.02% DDM in sodium phosphate buffer (pH 6.0) was used to equilibrate the column for elution. The receptor was then eluted from the rho-1D4-column with 0.02% DDM in sodium phosphate buffer (pH 6.0) supplemented with a nonapeptide (100 μM) whose sequence corresponds to the carboxyl terminus of rhodopsin (TETSQVAPA).

For capture of Meta-I, the DDM of the solubilized rhodopsin was reduced to 0.02% w/v and subsequently exchanged for 0.02–0.05% w/v digitonin on the 1D4-Sepharose column[38]. Rhodopsin was eluted in 2 mM phosphate buffer (pH 7.0) containing digitonin (0.02–0.05% w/v) and 100 μM C-terminal nonapeptide. The pooled, eluted rhodopsin fractions were concentrated to a final volume of ~400 μl using Centricon devices with a 10 kDa molecular weight cutoff (Amicon, Bedford, MA), followed by further concentration under argon gas to a volume of ~100 μl. All buffers were prepared fresh before purification[38].

**Synthesis of ¹³C-labelled retinals.** The ¹³C-labelled retinals were synthesized using unlabelled and ¹³C-labelled small molecule precursors. ¹³C5, ¹³C18 retinal was prepared by condensation of 4-bromo-2-methyl-2-butane with 1,2-¹³C-ethyl acetoacetate, and conversion of the labelled 6-methyl-5-hepta-2-one to ¹³C5, ¹³C18 retinal using conventional methods[61]. ¹³C10, ¹³C11 retinal was synthesized by condensation of β-ionone with ¹³CH₃¹³CN, which yields labelled β-ionylideneacetonitrile after dehydration. The latter was converted to ¹³C10, ¹³C11 retinal[61]. ¹³C12, ¹³C20 retinal was synthesized using aldol condensation of 2,3-dimethyl-5-(2',6',6'-trimethyl-1'-cyclohexene-1'-yl)-2,4-pentadienal with 1,3-¹³C acetone and conversion to labelled retinal. ¹³C14, ¹³C15 retinal was prepared by condensation of 5,6-dimethyl-8-(2',6',6'-trimethyl-1'-cyclohexene-1'-yl)-3,5,7-octatriene-2-one with di-¹³C-labelled acetonitrile and transformation to labelled retinal by conventional methods[61]. ¹³C18, ¹³C20 retinal was prepared by condensation of citral with 1,3-di-¹³C acetone and conversion of ketone product to labelled retinal[61].

**Solid-state NMR spectroscopy.** Solid-state NMR experiments were conducted at static field strengths of either 500 or 600 MHz using a 4 mm magic angle spinning (MAS) probe with a spinning rate of 10–12 kHz. NMR measurements were first made on rhodopsin in the dark at −83 °C. For Meta- I and -II, the NMR MAS rotor containing rhodopsin was ejected from the NMR probe, the NMR cap on the rotor was removed and the sample was illuminated for 1–2 min using a 400 W lamp with a 495 nm long-pass filter at room temperature (Meta-II)[62] or 4 °C (Meta-I)[38]. The cap was then replaced and the rotor inserted into a pre-cooled NMR probe where the sample temperature was able to reach −83 °C within ~5 min. We estimate that the conversion from rhodopsin to Meta-I in digitonin is >90% on the basis of ultraviolet/visible absorption and NMR spectra[38]. After conversion, we estimate that there is a loss of <10% of the Meta-I intermediate to Meta-II and opsin before the sample is cooled to −83 °C for NMR measurements[38]. We estimate that the conversion from rhodopsin to Meta-II in DDM is >90% and the loss of Meta-II to opsin is <5% before the sample is cooled to −83 °C (ref. 62).

The build-up curves were obtained by collecting DARR NMR spectra as a function of the mixing time during which magnetization is exchanged. The known curves were derived from measurements within rhodopsin at fixed distances (for example, retinal C5-C18, 1.4 Å; C8-C19, C12-C20, 2.4 Å; Cys110 Cβ-Cys187 Cβ, 3.6 Å; Cys187 Cβ-Gly188 Cα, 4.6 Å; Cys187 Cβ-Gly188 C=O, 5.3 Å). Spin diffusion is limited by the sparse labelling schemes that are typically used. The NMR assignments were based on mutation and/or mapping out specific correlations with unique resonances that have previously been assigned[18,19,34,38,63,64].

All ¹³C solid-state MAS NMR spectra were externally referenced to the ¹³C resonance of neat TMS at 0 p.p.m. at room temperature. Using TMS as the external reference, we calibrated the carbonyl resonance of solid glycine at 176.46 p.p.m. The chemical shift difference between ¹³C of DSS in D₂O relative to neat TMS is 2.01 p.p.m. The spectra shown were generally repeated at least twice on samples purified from different cell growths without deviations of more than ±0.2 p.p.m. in the reported chemical shifts.

**FTIR spectroscopy.** FTIR difference spectroscopy was performed with a Bruker Vertex 70 spectrometer with a mercury–cadmium–telluride detector[22–24]. Spectra were recorded by using time-resolved rapid-scan FTIR methods with a spectral resolution of 4 cm⁻¹. The wild-type and mutant rhodopsin samples were purified from HEK293S cell membranes in DDM and reconstituted at a 1:200 molar ratio into egg phosphocholine using biobeads for detergent removal. The results shown in Fig. 4 are for samples in phosphocholine membranes obtained at 0 °C. Samples

were prepared by drying solutions of rhodopsin between two CaF2 windows and then pre-equilibrating the sample with buffer (200 mM Bis-Tris propane or MES at pH 5.0 and 5.5) at the appropriate pH. Photolysis was carried out using an LED array centred at 530 nm for 1 s, and experiments were performed with an acquisition time of 12 s.

The conditions for NMR and FTIR are different. For NMR, the analysis relies on complete conversion (>90%) to Meta-I or Meta-II, which is facilitated by solubilization in digitonin or DDM, respectively. Rhodopsin is monomeric in DDM and is able to activate the G protein transducin[65]. Digitonin is unusual in that its hydrophobic end is composed of a rigid spirostan steroid moiety rather than flexible fatty acyl chains. The rigid framework effectively blocks the transition from Meta-I to Meta-II[66]. For FTIR, the analysis uses difference methods in which the FTIR spectrum of Meta-I or Meta-II is subtracted from the spectrum of rhodopsin. Only the vibrations that change in frequency or intensity contribute to the difference spectrum. Changes in pH or temperature can be used to shift the equilibrium between Meta-I and Meta-II. At low temperatures (below ~10 °C), the Meta-I ⇔ Meta-II equilibrium reflects a two-state transition in both DDM and phosphocholine bilayers, which breaks down into a series of Meta-II substates at higher temperature[22,50]. In both DDM and phosphocholine bilayers the conversion to Meta-IIbH+ happens rapidly (millisecond–second timescale) at 20 °C (refs 22,50), which requires the use of time-resolved methods to follow the transition. For NMR, we convert fully to Meta-II at room temperature, but require several minutes to low-temperature trap the Meta-II intermediate, which we assume is predominantly Meta-IIbH+, before it decays.

**Data availability.** NMR data that support the findings of this study have been deposited in the BMRB databank with the accession codes 26812 (rhodopsin) and 26813 (Meta-II) (http://www.bmrb.wisc.edu). The coordinates of Meta-II from the guided molecular dynamics simulations and remaining data are available on reasonable request from the corresponding author.

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

## Acknowledgements

This work was supported by a grant from the NIH (GM 41412) to S.O.S.

## Author contributions

S.O.S. and P.J.R. conceived and designed the work; N.K., A.P. and M.E. prepared isotopically labelled rhodopsin; C.A.O. and P.J.R. prepared rhodopsin mutants; A.H. and M.S. synthesized isotopically labelled retinals; N.K., M.E. and M.Z. collected and analysed NMR data; E.Z. and R.V. collected and analysed FTIR data.

## Additional information

**Competing financial interests:** The authors declare no competing financial interests.

**How to cite this article**: Kimata, N. *et al.* Retinal orientation and interactions in rhodopsin reveal a two-stage trigger mechanism for activation. *Nat. Commun.* 7:12683 doi: 10.1038/ncomms12683 (2016).

