## [Peer Review File · Nature Communications]

Reviewer #1 (Remarks to the Author):

This is a very interesting and experimentally solid work from the experts in the field of visual rhodopsin activation. It powerfully combines forces of the experts in GPCR expression, retinal analogs, solid-state NMR, and FTIR spectroscopy of photoreceptor proteins. The paper presents new experimental results in four categories. First, there are new and revised results on the proximity of retinal methyl groups and selected ring atoms to a number of opsin residues (Tyr, Phe, Met, His, Gly), both in the dark and Meta II states, as obtained by DARR experiments. This nicely addresses the recent controversy on the orientation of retinal in Meta II. Second, there are new NMR results on the light-induced rearrangement of the complex of residues on the extracellular side of retinal (Tyr268, Gly188, Cys187, Tyr191, Met288, Ser186), both in Meta I and Meta II intermediates. Third, there are FTIR data showing the effects of some of the above residues on Meta I/Meta II equilibria. Fourth, the supplementary file has very interesting data on the reestablishment of Tyr191/Ret contacts upon Meta II decay, which adds to the arguments regarding the true orientation of retinal in Meta II. Finally, the authors tried to integrate all of these, and many of the previous, results into a general hypothesis of a two-stage rhodopsin activation.

While the experimental results are very solid and interesting, and must be published, this reviewer believes that the integrated hypothesis of a two-stage rhodopsin activation is not presented convincingly. While the proposed mechanism may be 100% correct, the presentation of this mechanism, and the way it integrates the experimental data with the vast body of the previous knowledge, need to be improved. In the present form, it is inaccessible to anyone not in the immediate narrow field, the abstract and the discussion look disconnected from the experimental results, and the whole paper looks somewhat disjointed. More specifically, the following points will be unclear for most readers: i) why the retinal proximity data are collected on Meta II only (not on Meta I), but used to argue for the stage I of the triggering; ii) why and how the NMR data collected just for a few residues are used to build much more general mechanism. In other words, the experimental results should be put into the context of what is already known much more explicitly, and the new and old results should be contrasted better; iii) the FTIR data are not integrated well with the NMR data. This piece looks foreign, unless the integration is explained better. Additionally, better description of what was actually done is needed. Not a single FTIR spectrum is shown. iv) Methods description is very incomplete. For example, the conditions for Meta I are not given anywhere, chemical shifts for slices are often missing, FTIR reference spectra are not there, etc.; v) Similarly, many statements (which may look obvious to the authors, but not to most readers) are not supported by references. More specific examples of these points are listed below. In my opinion, the paper would be much stronger if the focus were shifted from the "high level integration" two stage hypothesis to the discussion of the true orientation of retinal in Meta II in the context of associated protein changes. On the other hand, if the focus remains the same, the hypothesis should be much better integrated with the presented results and placed in context of the existing knowledge.

Specific points, some of which are just minor editorial issues and some illustrate the general points mentioned above:

- 1) p. 1, address #1 is incomplete
- 2) p. 2. abstract - see above. The main point that the proposed hypothesis integrates a lot of the old data with some of the new data presented in the paper is missing. A better placement into the context would be helpful.
- 3) p. 3, "As a result, it has been a challenge" - the logic is not clear.
- 4) p. 5, "The NMR and FTIR approaches make use of low temperature to trap the active Meta-II state" - to be fair, one can argue that NMR experiments are done on the frozen detergent micelles, which is not completely native. On the other hand, it seems that FTIR results were obtained on the HEK cell membranes (even though the methodology description is not very clear here). The difference in sample conditions between NMR and FTIR is not discussed.

- 5) p. 7, "the relative intensity of the cross peak to the Phe261" - relative to what?
- 6) p. 7, reference to Fig. S2, actually refers to Fig. S2B. The data presented in Fig. S2A are never discussed or referred to in the main text.
- 7) p. 7, "lost, consistent with an increased separation between these tyrosines and the retinal C19 carbon" - an alternative explanation would be retinal rotation, which should be mentioned and argued against once more.
- 8) p. 7, "loss of intensity of these tyrosines with the retinal C19 methyl resonance" - style
- 9) p. 8, "The row through the C20 diagonal resonance yields" - here and elsewhere, it should be clearly stated that this resonance may change between Rd and Meta II and the actual positions indicated in the figures.
- 10) p. 9, "Tyr2686.51 has the highest subfamily conservation" - elsewhere in the paper it says "one of the highest", also what about W265?
- 11) p. 11, "This cross peak does not lose intensity as previously assigned" - style
- 12) p. 11, the discussion of Fig. 2 peak intensity changes needs to be more quantitative. E.g., what is the intensity increase for the 268/122 crosspeak?
- 13) p. 11, "Both increase to $> 6 \text{ \AA}$ " - style
- 14) p. 12, "Tyr2686.51 on TM helix H6 is strongly hydrogen bonded to Glu181" - indicate according to which structure
- 15) p. 12, "and that Tyr191EL2 has shifted relative to Glu181EL2" - not clear where this came from, as E181 has not been observed in this paper. Needs better explanation.
- 16) p. 13, "against Ala2726.55 H6" - style
- 17) p. 13, "the inward tilt of H6" - confusing, as there is also outward tilt and rotation of H6 mentioned elsewhere. Would be nice to specify which half of the helix is involved for clarity.
- 18) p. 13, it would be nice to have a schematics of the proposed changes in the EL2 cluster
- 19) p. 14, "The observation that Tyr191EL2 and Tyr2686.51 strongly stabilize the Meta-I state suggests that in crystal structures of opsin and/or Meta-II, the inactive hydrogen-bonding network reforms in this region of the receptor upon the decay of Meta-II to opsin" - needs better explanation
- 20) p. 15, the first subsection of the discussion is not needed as a separate section, in my view, it mainly repeats introduction.
- 21) p. 15, "mechanism is highlighted by the $\sim 35 \text{ kcal/mol}$ " - not clear
- 22) p. 15, "Our current results show this change is mediated by interactions of the retinal with Phe2616.44 and Trp2656.48" - not clear as W265 has not been detected in this work.
- 23) p. 16, discussion on the interaction of His211 and Glu122 - once again, it is not clear where this comes from. Glu122 has not been measured in this work, if this is from the old data, the references should be given.
- 24) p. 16, "results in motion of the retinal PSB proton away from the stabilizing interaction with its counter-ion Glu113" - a bit unorthodox, as normally people speak about the PSB counterion rather than proton counterion.
- 25) p. 17, "Upon isomerization, the C19 methyl group rotates in the opposite direction" - reference is needed
- 26) p. 19, " ^{13}C label rhodopsin" - style
- 27) Fig. 1 legend - PDB accession should be given; positions of the diagonal resonances used should be indicated; "scaled to the C12-C20 cross peaks" - specify how;
- 28) Fig. 1H - Signal-to-noise is poor resulting in some negative bands (phasing?). May be comment on that?
- 29) Fig. 3 legend - "are shown of rhodopsin minus Meta-I (orange) and rhodopsin minus Meta-II (black) of rhodopsin" - style
- 30) Fig. 3 - large perturbations of Met and Tyr other than those discussed are observed, but ignored. Assignments of those resonances are not explained (reference if from the previous work)
- 31) Fig. 4 legend refers to methods for the reference FTIR spectra, but those are not described there.
- 32) Fig. 3 - two panels are labeled C
- 33) Suppl. p. 4 - "the $3\text{C}\epsilon$ - Met"
- 34) Suppl. p. 5 - for the discussion of the ring inversion and other possible changes in retinal

(resulting in C18-Phe261 proximity) a figure would be nice

35) Suppl. Fig. S4 legend and the following paragraph overlap significantly. May be merge the two.

36) Suppl. p. 9 - two references to non-existing Fig. 3E are made.

37) Suppl. p. 12 - "absence of a cross peak between the retinal chromophore and ^{13}C -labeled tyrosine in Meta-II" - not true in general, should specifically name the retinal part discussed

38) Suppl. Fig. S6 - normalization of the C5-C18 cross peak is confusing, as its amplitude decreases

39) Suppl. p. 13 - (S.O. Smith, personal communication) - not sure one can refer to personal communication from one of the authors. May be unpublished is more appropriate.

40) Suppl. p. 13 - "The high frequency of the ^{13}C 5 chemical shift suggests that the retinal with the flipped ring orientation is bound to Lys296 as a protonated Schiff's base" - explain and reference

41) Suppl. p. 14 - I would move some of the very interesting discussion on Meta III into the main text.

42) Suppl. Fig. 7 - the legend needs tightening, it is not very clear and accessible.

Reviewer #2 (Remarks to the Author):

This manuscript reports the results of a very informative, if not groundbreaking, study on vertebrate rhodopsin. SS-NMR dipolar assisted rotation resonance (DARR) along with ^{13}C isotope labeling of the retinal at specific positions and/or phenylalanine and tyrosine was used to measure distances between specific atoms in the protein. Although this method cannot pick up cross peaks when distances are more than 6-6.5 Å, absence of such peaks along with signal strength when detected can be used to infer the proximity of specific groups, especially when comparing the structure of the dark state of rhodopsin with bleaching intermediates.

One major conclusion concerns the orientation of the β -ionone ring in the dark-state and Meta-II intermediate. While the activated state structure has been elucidated for opsin it suffers from the lack of a retinal chromophore thus precluding direct visualization of how the retinal isomerization triggers the conversion to the active state. In two structures of Meta-II obtained using crystals of the M257Y mutant and opsin containing soaked-in retinal, the orientation of the ring has flipped from the dark-state. However, the DARR experiment, which measures the low-temperature trapped active state of rhodopsin photointermediate shows convincingly that the β -ionone ring does not flip under these conditions.

The paper also provides detailed insight into the molecular events which lead to the transition to Meta II including outward rotation of helix-6, a key feature of GPCRs, and conveyance of a signal to EL2. This includes new information on the interaction of Tyr268 on helix-6 with Tyr191 (β -4 strand on EL2) and Glu181 (β -3 strand of EL2 which in the dark state interacts with the SB). An FTIR pH difference technique developed by the Vogel group to study the transition from Meta-I to Meta-II (and sub-states) also provides additional support for some of the conclusions.

Overall, this is an innovative, technically sound study which provides important new insights into structural changes leading to rhodopsin activation, the paper will be of wide interest and will stimulate studies of other ligand-activated GPCRs.

There are a number of minor problems with the manuscript which the authors need to correct before publication:

Figure 1 caption is confusing: Does Figure 1B show crystal structure of dark state or Meta-II intermediate. Also cytoplasmic and exterior membrane surface should be labeled in Figure 1B. Note that Tyr191 does not appear to be the closest residue to retinal C18. If Figure 1B is the dark

state then this is understandable but in that case one of the Meta II forms of the crystal structure should also be shown.

Page 6: It should be noted that the referred to peak at 21.6 ppm is not shown.

Figure 1C: In bottom row of DARR data add lines to show the Phe261-C18 cross peaks as done for DARR row above.

Page 7, paragraph 2: Thr118 is not shown in Figure 1B but mentioned in text. The authors should consider adding this to Figure.

Figure 2B: The label Ile189 is partially clipped at top of font.

Figure 3: Crystal structure is mislabeled C and not D.

Page 12, second paragraph: Refer to Figure 3D (now 3C).

Figure 4: It would be easier to compare these plots if the pKas (inflection points) were indicated on the curves. This is especially true since x-axis pH scale is not reproduced on top set of panels.

Discussion: An earlier study which involved FTIR difference spectroscopy of isotope labeled rhodopsin suggested a role for Tyr268 in rhodopsin activation as well as helix-6 reorientation. This work should be mentioned and referenced to in Discussion (DeLange, et al. (1998) Tyrosine structural changes detected during the photoactivation of rhodopsin. *J Biol Chem* 273, 23735-23739).

Supplementary Figure S5: Panel D is not described. For example, which state does purple colored structure refer to?

Reviewer #3 (Remarks to the Author):

The authors present an extensive solid-state NMR study supported by FTIR and mutagenesis on rhodopsin. The aim of their work is to show that the activation of rhodopsin requires multiple steps of interaction between the retinal chromophore and opsin after the isomerisation event. Recent X-ray studies have shown differently oriented retinal co-factors. Solid-state NMR is here the method of choice to obtain specific data to resolve this issue.

Isotope labelled retinal was incorporated into residue-selectively labelled opsin. The observation of dipolar contacts between the ^{13}C spins enabled the authors to conclude details on the retinal orientation within the binding pocket after light activation and trapping the Meta-II state.

The authors suggest that in the first stage of the activation, the retinal beta-ionone ring triggers rotation of helix 6 and stabilizes Meta II. In the second step, C19 and C20 in the retinal polyene chain are suggested to sterically interact with Tyr191 and Tyr268, which also includes deprotonation of the retinal Schiff base.

Overall, data interpretation is sound and solid-state NMR is certainly the right approach. However, I could not follow how the authors derived a sequential step model, for which time-resolved data or at least trapping of batho, meta-I and meta-II would have been required.

Although I am familiar with the field, I would also recommend some alterations to the paper as it is difficult follow without consulting the literature and some data presentation appears selective:

The materials and method section is very short. Some more details would be helpful, especially, regarding the state of the samples. Was solid-state NMR applied to frozen detergent solution or to

proteoliposome samples? What is the source organism of the rhodopsin used here?

It would be helpful to include some of the 2D spectra leading to the data in Fig. 1. The authors only show 1D slices of the spectra.

Fig. 1G: The authors mention a comparison to build-up curves from model compounds but it is not clear whether these data are shown here. Furthermore, spin-diffusion build-up curves can be difficult to quantify. The authors seem to present here calculated curves for certain distances. Further details should be provided.

Some explanations should be provided how the MI/MII trapping was performed and how good the trapping efficiency was.

It should be somewhere summarized how the assignment was achieved.

Reviewer 1

The first reviewer found the “*manuscript to be experimentally solid*” and to “*powerfully combine forces of the experts in GPCR expression, retinal analogs, solid-state NMR, and FTIR spectroscopy of photoreceptor proteins*”.

The reviewer raised 5 major points and 42 additional points that we have addressed below.

Point 1. While the experimental results are very solid and interesting, and must be published, this reviewer believes that the integrated hypothesis of a two-stage rhodopsin activation is not presented convincingly. While the proposed mechanism may be 100% correct, the presentation of this mechanism, and the way it integrates the experimental data with the vast body of the previous knowledge, need to be improved. In the present form, it is inaccessible to anyone not in the immediate narrow field, the abstract and the discussion look disconnected from the experimental results, and the whole paper looks somewhat disjointed.

In my opinion, the paper would be much stronger if the focus were shifted from the "high level integration" two-stage hypothesis to the discussion of the true orientation of retinal in Meta II in the context of associated protein changes. On the other hand, if the focus remains the same, the hypothesis should be much better integrated with the presented results and placed in context of the existing knowledge.

Response: The major concern of the reviewer is the presentation of the results. The manuscript was originally focused on defining the orientation of the retinal chromophore. However, in the course of these studies it became clear that the differences in orientation between NMR and crystallography have very different implications about the mechanism of activation, and the manuscript shifted toward a new focus.

To address the concern of the reviewer in terms of the presentation and access to the non-expert reader, we have revised the title, abstract, introduction and discussion.

Title: We have changed the title in order to highlight both the “retinal orientation” and the “activation mechanism” aspects of the manuscript. We believe the retinal orientation component will be of considerable interest to the visual receptor community, while the activation mechanism will have impact across the GPCR field where the details about how agonists trigger activation are lacking.

Abstract. We have revised the abstract as suggested by the reviewer to address the concern listed below that the proposed mechanism integrates old and new data.

“We integrate these observations with previous structural and functional studies to propose a two-component mechanism for rhodopsin activation.”

We have also explicitly indicated that the orientation of the retinal chromophore found in Meta II differs from that in the crystal structures.

“The orientation of the retinal differs from that in recent active-state rhodopsin crystal structures.”

Introduction. We have revised the introduction in several respects.

- a. We introduce one of the motivating questions behind the research, namely how retinal isomerization and deprotonation of the retinal PSB generate the large helix rearrangements on the intracellular side of the receptor. This question is raised in conjunction with the puzzling observation that 35 kcal/mol of light energy is stored in the primary photoproduct bathorhodopsin, yet very little changes on the extracellular side of the receptor as this energy is released upon formation of the active Metarhodopsin II intermediate.

Page 4. *“Other than the large change in retinal configuration and orientation, the crystal structures of active rhodopsin show almost no change in structure on the extracellular side of the receptor when compared to the large changes observed on the intracellular side¹³. This observation is surprising as a substantial amount of absorbed light energy (~35 kcal/mol¹⁴) is stored within retinal-protein interactions in the primary photoproduct bathorhodopsin and then released as the retinal and surrounding protein residues relax in the transition to the active Meta-II intermediate². The lack of structural changes in the residues surrounding the retinal raises the question of how retinal isomerization and deprotonation of the retinal PSB generate the large helix rearrangements on the intracellular side of the receptor.”*

- b. We state explicitly that the retinal orientations observed by crystallography and NMR imply different mechanisms for activation.

Page 5. *“The opposite orientations of the retinal chromophore observed in the different Meta-II structures suggest different mechanisms for activation. The orientation of the retinal within the active-state crystal structures argues that steric interactions resulting from retinal isomerization lead to an expansion of the retinal binding pocket to provide space for rotation of the bulky β -ionone portion of the retinal. That is, steric interactions are the dominant force driving the outward rotation of H6. In contrast, the orientation of the β -ionone ring end of the retinal relative to the extracellular surface does not change in the NMR structure of Meta-II¹⁸. Rather, the protonated Schiff base end of the chromophore (including the C20 methyl group) undergoes the largest rotation upon retinal isomerization^{16,17}, consistent with electrostatic interactions having the dominant contribution to receptor activation¹⁹.”*

Discussion. We have revised the discussion extensively as suggested in points 20-25 below, removed the subheadings as per journal style, and edited the text to highlight the importance of these results in understanding the general mechanism of GPCR activation.

Point 2. More specifically, the following points will be unclear for most readers: i) why the retinal proximity data are collected on Meta II only (not on Meta I), but used to argue for the stage I of the triggering and ii) why and how the NMR data collected just for a few residues are used to build much more general mechanism. In other words, the experimental results should be put into the context of what is already known much more explicitly, and the new and old results should be contrasted better.

Response: We have revised the concluding paragraph in the introduction to more clearly indicate how we use NMR and FTIR to probe Meta I and Meta II. We have also revised the end of the first paragraph in the Results section to describe how select NMR distance measurements are combined with previous studies in the literature to propose a general mechanism of activation.

Pages 5 (Introduction). *“In refining the orientation of the all-trans retinal in Meta-II based on solid-state NMR measurements, we propose an activation mechanism that builds on previous studies by emphasizing the changes in extracellular residues in close proximity to the retinal. We are able to follow changes in these residues by comparing differences between rhodopsin and Meta-II, or in some cases, between rhodopsin and Meta-I, which precedes Meta-II in the photoreaction pathway. To address how specific residues in close association with the retinal control the equilibrium between Meta-I and Meta-II (i.e. the final step in the reaction pathway), we take advantage of FTIR spectroscopy²⁰. FTIR difference spectroscopy provides a complementary approach to NMR for characterizing the contribution of hydrogen-bonding interactions of specific amino acids to receptor activation²⁰⁻²². The mechanism that emerges from these studies implicates steric interactions as the dominant force driving structural changes between rhodopsin and Meta-I, while electrostatic (and hydrogen-bonding) interactions control the formation of the active Meta-II conformation.”*

Page 6 (Results). *“Below, we use NMR to measure a few specific distances between the retinal and surrounding protein to define the orientation of the retinal in Meta II. These constraints along with the results from previous biophysical and biochemical studies are used to suggest a general mechanism of receptor activation.”*

Point 3. The FTIR data are not integrated well with the NMR data. This piece looks foreign, unless the integration is explained better. Additionally, better description of what was actually done is needed. Not a single FTIR spectrum is shown.

Response: We have revised the Introduction as described in the response to Point 2. We have completely written the last section of the Results that contains the FTIR data, and we have extensively revised the Supporting Information (Supplementary Figure 7) and Methods sections. We now show in Supplementary Figure 7 the full FTIR spectra as a function of pH for wild-type rhodopsin and the Y268F mutant.

Point 3. Methods description is very incomplete. For example, the conditions for Meta I are not given anywhere, chemical shifts for slices are often missing, FTIR reference spectra are not there.

Response: We have expanded the Methods and Supporting Information sections. The Meta I conditions are provided in Methods and the FTIR reference spectra are now shown in Supplementary Figure 7. We have explicitly added the chemical shift values for the rows taken from the 2D NMR spectra.

Figure Legend 1. *“The rows selected correspond to the diagonal chemical shifts of the $^{13}\text{C}18$ resonance in rhodopsin at 21.6 ppm and Meta-II at 20.9 ppm, which maximize the crosspeak intensities.*

Rows through the $^{13}\text{C}19$ diagonal resonances in rhodopsin (black) at 14.7 ppm and Meta-II (red) at 13.8 ppm obtained with the receptor regenerated with $^{13}\text{C}8$, $^{13}\text{C}19$ retinal and incorporating ^{13}C -ring Phe and $^{13}\text{C}\zeta$ -labeled Tyr.

Rows through the $^{13}\text{C}20$ diagonal resonances in rhodopsin (black) at 16.4 ppm and Meta-II (red) at 13.7 ppm obtained with the receptor regenerated with $^{13}\text{C}12$, $^{13}\text{C}20$ retinal and incorporating $^{13}\text{C}\zeta$ -labeled Tyr.

Rows are taken through the $^{13}\text{C}\zeta$ -Tyr diagonal resonance at 155.2 ppm in rhodopsin and 156.1 ppm in Meta II.”

Supplementary Figure 5. *“Rows are taken through the Met86 diagonal resonance at 13.6 ppm in rhodopsin (black) and 15.0 ppm in Meta II (red).”*

“In (b), the rows are taken through the $^{13}\text{C}20$ diagonal resonance at 16.4 ppm in rhodopsin and 13.7 ppm in Meta II.”

“(c) We present a row through the retinal $^{13}\text{C}20$ resonance at 13.7 ppm of a DARR spectrum of Meta-II labeled with $^{13}\text{C}\zeta$ -tyrosine, $^{13}\text{C}\alpha$ -glycine and containing $^{13}\text{C}12$, $^{13}\text{C}20$ -retinal.”

Supplementary Figure 6. *“Panel (a) presents a row through the diagonal resonance of $^{13}\text{C}\epsilon$ -Met288^{7,35} at 17.2 ppm in dark state rhodopsin showing cross peaks with both the tyrosine $^{13}\text{C}\zeta$ and $^{13}\text{C}=\text{O}$ carbons.*

(b) DARR NMR of Meta II. Tyr191^{EL2} and Tyr192^{EL2} generate the strongest cross peaks to the $^{13}\text{C}\epsilon$ -Met288^{7,35} diagonal resonance at 12.2 ppm.”

Point 4. Similarly, many statements (which may look obvious to the authors, but not to most readers) are not supported by references. More specific examples of these points are listed below.

Response: We have inserted additional references throughout the main text and Supplementary Information to better integrate our studies with the vast literature on rhodopsin structure and mechanism.

Specific points, some of which are just minor editorial issues and some illustrate the general points mentioned above:

1) p. 1, address #1 is incomplete

Response: Corrected.

2) p. 2. abstract - see above. The main point that the proposed hypothesis integrates a lot of the old data with some of the new data presented in the paper is missing. A better placement into the context would be helpful.

Response: We have revised the abstract accordingly.

Abstract. *"We integrate these observations with previous structural and functional studies to propose a two-component mechanism for rhodopsin activation."*

3) p. 3, "As a result, it has been a challenge" - the logic is not clear.

Response: We have removed this sentence in the process of improving the flow and logic in the introduction.

4) p. 5, "The NMR and FTIR approaches make use of low temperature to trap the active Meta-II state" - to be fair, one can argue that NMR experiments are done on the frozen detergent micelles, which is not completely native. On the other hand, it seems that FTIR results were obtained on the HEK cell membranes (even though the methodology description is not very clear here). The difference in sample conditions between NMR and FTIR is not discussed.

Response: We now include a section in the expanded Methods to describe the differences in sample preparation and experimental conditions.

Pages 20-21. *"The wild-type and mutant rhodopsin samples were purified from HEK293S cell membranes, purified in DDM and reconstituted at a 1:200 molar ratio into egg phosphocholine (PC) using biobeads for detergent removal. The results shown in Fig. 4 are for samples in PC membranes obtained at 0 °C. Samples were prepared as sandwich samples with 200 mM BTP buffer (MES at pH 5.0 and 5.5) including pre-equilibration. Photolysis was carried out using an LED array centered at 530 nm for 1s, and experiments were performed with an acquisition time of 12 s."*

The conditions for NMR and FTIR are different. For NMR, the analysis relies on complete conversion (>90%) to Meta-I or Meta-II, which is facilitated by solubilization in digitonin or DDM, respectively. Rhodopsin is monomeric in DDM and is able to activate the G protein transducin⁵⁸. Digitonin is unusual in that its hydrophobic end is composed of a rigid spirostan steroid moiety rather than flexible fatty acyl chains. The rigid framework effectively blocks the transition from Meta-I to Meta-II⁵⁹. For FTIR, the analysis uses difference methods, which allows one to easily shift the equilibrium between Meta-I and Meta-II by pH or temperature. At low temperatures (below ~10 °C), the Meta-I ⇌ Meta-II equilibrium reflects a two-state transition in both DDM and PC bilayers, which breaks down into a series of Meta-II substates at higher temperature^{20,43}. In both DDM and PC bilayers the conversion to Meta-IIbH+ happens rapidly (millisecond-second time scale) at 20 °C^{20,43}, which requires the use of time-resolved methods to follow the transition. For NMR, we convert fully to Meta-II at room temperature, but require several minutes to low-temperature trap the Meta-II intermediate, which we assume is predominately Meta-IIbH+, before it decays."

5) p. 7, "the relative intensity of the cross peak to the Phe261" - relative to what?

Response: We revised this sentence.

“However, the intensity of the cross peak to the Phe261^{6.44} ring ¹³C resonances remains approximately the same as in rhodopsin indicating that the ring does not change orientation (i.e. flip) in the conversion to Meta-II.”

6) p. 7, reference to Fig. S2, actually refers to Fig. S2B. The data presented in Fig. S2A are never discussed or referred to in the main text.

Response: We have revised this sentence accordingly.

Page 7. “The position of the β -ionone ring is additionally constrained by contacts between the ¹³C5, ¹³C18 and the ¹³C16, ¹³C17 retinal resonances and residues (Met207^{5.42} and His211^{5.46}) on H5 (Supplementary Fig. 3). Furthermore, in the Meta-II-opsin and Meta-II-M257Y crystal structures, the rotation of the β -ionone ring is predicted to bring the C18 methyl group to within the DARR distance limit (~ 6 Å) of ¹³C ξ -Tyr191^{EL2}, which is not observed (Fig 1d).”

7) p. 7, "lost, consistent with an increased separation between these tyrosines and the retinal C19 carbon" - an alternative explanation would be retinal rotation, which should be mentioned and argued against once more.

Response: We have added a sentence at this point in the text.

Page 8. “Moreover, we show that the alternative explanation, the flip of the β -ionone ring, does not occur until Meta-II decays (Supplementary Fig. 4).”

8) p. 7, "loss of intensity of these tyrosines with the retinal C19 methyl resonance" – style

Response: The sentence has been revised.

Page 8. “Below, we now show that the chemical shifts of both Tyr191 and Tyr268 change in Meta-II contributing to the loss of crosspeak intensity between the ¹³C ξ -Tyr191^{EL2}/Tyr268^{6.51} resonances and the retinal C19 methyl resonance. Moreover, we show that the alternative explanation, the flip of the β -ionone ring, does not occur until Meta-II decays (Supplementary Fig. 4).”

9) p. 8, "The row through the C20 diagonal resonance yields" - here and elsewhere, it should be clearly stated that this resonance may change between Rd and Meta II and the actual positions indicated in the figures.

Response: We have added an explicit statement on Figure Legend 1 (where we introduce how we select rows from the 2D spectra) and have added a new Supplementary Figure 2 in which we describe the 2D NMR experiment and how rows are selected.

Figure 1 Legend. “Since the chemical shifts can change between rhodopsin and Meta II, the rows selected correspond to the diagonal chemical shifts of the ¹³C18 resonance in rhodopsin at 21.6 ppm and Meta-II at 20.9 ppm. These rows maximize the crosspeak intensities.”

10) p. 9, "Tyr268^{6.51} has the highest subfamily conservation" - elsewhere in the paper it says "one of the highest", also what about W265?

Response: Tyr268^{6.51} is the second highest. Lys296 is the highest. Trp265 is highly conserved across the family A GPCRs and not just in the opsin subfamily. We added a sentence to make this point about subfamily vs family conservation clearer.

Page 10. *"Tyr268^{6.51} has the highest subfamily conservation (97%) in the visual GPCRs after Lys296^{7.43}, the site of retinal attachment, indicating that its position and interactions are critically important within the visual receptors."*

11) p. 11, "This cross peak does not lose intensity as previously assigned" - style

Response: The description of the cross peak intensity changes on page 11 has been rewritten and this sentence has been deleted.

12) p. 11, the discussion of Fig. 2 peak intensity changes needs to be more quantitative. E.g., what is the intensity increase for the 268/122 crosspeak?

Response: We have added selected rows from the contour plots in Figure 2 to the figure in order to clearly show the intensity changes. The most relevant rows correspond to the Tyr-Gly cross peaks. (We do not have a 268/122 cross peak; rather the cross peak intensity that the reviewer is referring to is likely 268/188).

We have modified the figure legend to discuss the intensity differences in these rows.

Page 30: *"Above panel (a) are shown rows through the Tyr-Gly crosspeaks. The rows better illustrate the intensity change occurring in the Tyr268^{6.51}-Gly188^{EL2} peak upon activation. The observation that the Tyr178^{EL2}-Gly114^{3.29} does not change intensity is consistent with the lack of influence of the Y178F mutation upon the Meta-I – Meta-II transition (see Fig. 4g)."*

13) p. 11, "Both increase to > 6 Å" - style

Response: The sentence has been revised.

Page 11. *"Both Tyr-Cys distances increase to >6 Å in the opsin crystal structure⁴."*

14) p. 12, "Tyr268^{6.51} on TM helix H6 is strongly hydrogen bonded to Glu181" - indicate according to which structure

Response: In the supporting Tables, we have listed the crystal structure distances that are likely to be of interest to the reader. The Tyr268-Glu181 distances were not previously included. We have now listed the O...O distances between the Tyr268 side chain and the Glu181 carboxyl side chain in Table S5 and made a note of this in the main text.

Page 12. *"Tyr268^{6.51} on TM helix H6 is strongly hydrogen-bonded to Glu181^{EL2} on the β3 strand of EL2 and to Tyr191^{EL2} on the β4 strand of EL2 (Fig. 3d, Supplementary Table 5)."*

15) p. 12, "and that Tyr191EL2 has shifted relative to Glu181EL2" - not clear where this came from, as E181 has not been observed in this paper. Needs better explanation.

Response: This sentence has been revised.

Page 12. *"The NMR data discussed above show that Tyr268^{6.51} has shifted relative to the retinal in Meta-II, while the large downfield chemical shift of Tyr191^{EL2} suggests that it has shifted relative to Glu181^{EL2}."*

16) p. 13, "against Ala272^{6.55} H6" - style

Response: The sentence has been revised.

Page 13. *"...packed against Ala272^{6.55} on H6"*

17) p. 13, "the inward tilt of H6" - confusing, as there is also outward tilt and rotation of H6 mentioned elsewhere. Would be nice to specify which half of the helix is involved for clarity.

Response: The sentence has been revised.

Page 13. *"Tyr191^{EL2} is tightly packed against Ala272^{6.55} on H6, one helical turn from Tyr268^{6.51}, and its motion toward Glu181^{EL2} would release steric constraints hindering the inward tilt of the extracellular end of H6."*

18) p. 13, it would be nice to have a schematics of the proposed changes in the EL2 cluster

Response: The schematic in Figure 5 was intended for this purpose. We have included additional views of the region containing Met288, Tyr191, Tyr268 and Glu181 in Supplementary Figure 6 that should help visualize the changes.

19) p. 14, "The observation that Tyr191^{EL2} and Tyr268^{6.51} strongly stabilize the Meta-I state suggests that in crystal structures of opsin and/or Meta-II, the inactive hydrogen-bonding network reforms in this region of the receptor upon the decay of Meta-II to opsin" - needs better explanation

Response: This sentence has been moved to the Discussion section and revised.

Pages 18. *"The net effect of PSB deprotonation and rearrangement of the hydrogen-bonding network involving EL2 is a shift of Tyr191^{EL2} away from H6. We propose that this motion allows the extracellular end of H6 to pivot inward. In the visual pigments, Tyr191^{EL2} has a high level of sequence identity (61%), and an overall level of conservation of 83% as either tyrosine or tryptophan. Interestingly, even though Tyr191^{EL250} or Tyr268^{6.51} contribute to Meta-I stability (Fig. 4), mutation of either these residues results in a substantial drop in G protein activation^{38,50}. This dual influence of Tyr191^{EL2} and Tyr268^{6.51} on Meta-I stability and Meta-II activity is consistent with one set of hydrogen-bonding interactions stabilizing Meta-I and a second set stabilizing Meta-II (Fig. 5). The observation that Tyr191^{EL2} and Tyr268^{6.51} strongly stabilize the Meta-I state suggests that in crystal structures of opsin and/or Meta-II, the inactive-state hydrogen-bonding network reforms in this region under the conditions used for crystallization."*

20) p. 15, the first subsection of the discussion is not needed as a separate section, in my view, it mainly repeats introduction.

Response: We have largely removed this section in the Discussion.

21) p. 15, "mechanism is highlighted by the ~35 kcal/mol" - not clear

Response: We have brought this discussion to the introduction and elaborated on the fact that 35 kcal/mol is a substantial amount of energy.

Page 4. *“Other than the large change in retinal configuration and orientation, the crystal structures of active rhodopsin show almost no change in structure on the extracellular side of the receptor when compared to the large changes observed on the intracellular side¹³. This observation is surprising as a substantial amount of absorbed light energy (~35 kcal/mol¹⁴) is stored within retinal-protein interactions in the primary photoproduct bathorhodopsin and then released as the retinal and surrounding protein residues relax in the transition to the active Meta-II intermediate². The lack of structural changes in the residues surrounding the retinal raises the question of how retinal isomerization and deprotonation of the retinal PSB generate the large helix rearrangements on the intracellular side of the receptor.”*

22) p. 15, "Our current results show this change is mediated by interactions of the retinal with Phe2616.44 and Trp2656.48" - not clear as W265 has not been detected in this work.

Response: This section has been revised.

Page 15. *“NMR³² and fluorescence³³ studies both reveal that the first major change in receptor conformation is rotation of H6 in the formation of Meta-I, which breaks the intracellular Arg135^{3.50}-Glu247^{6.30} ionic lock³⁴ and results in Trp265^{6.48} displacement^{32,35}. These changes are accompanied by small changes at the cytoplasmic end of H5^{33,36}. Our current results indicate direct contact of the β -ionone ring with His211^{5.46} and Phe261^{6.44} in Meta-II. Previously, we reported that activation results in a loss of the retinal C18-Trp265^{6.48} contact¹⁷. Mutation of either Phe261^{6.44} or Trp265^{6.48} lowers the initial rate of G protein activation, ~20-60% for F261A rhodopsin³⁷ and ~90% in W265F rhodopsin³⁸. The steric contact of the β -ionone ring with Phe261^{6.44} observed in Meta-II suggests that the phenylalanine ring acts a lever for rotation of H6. This residue is part of a transmission switch in the conserved TM core of GPCRs^{13,39}.”*

23) p. 16, discussion on the interaction of His211 and Glu122 - once again, it is not clear where this comes from. Glu122 has not been measured in this work, if this is from the old data, the references should be given.

Response: This section has been revised.

Page 15-16. *“The conserved proline (Pro215^{5.50}) on H5 is also a component of the transmission switch and results in a free backbone carbonyl at His211^{5.46}. This carbonyl forms an interhelical hydrogen-bond with Glu122^{3.37} in rhodopsin, which is replaced in Meta II by a direct interaction between the His211^{5.46} and Glu122^{3.37} side chains⁹. The strong steric contacts of the β -ionone ring with His211^{5.46} and Glu122^{3.37} that drive this transition allow H5 to reorient in Meta-II. Mutation of Glu122^{3.37} breaks their direct interaction and destabilizes Meta-II⁴⁰. Retinal analogs in which the ring is truncated shift the conformational equilibrium between the Meta-I and Meta-II intermediates toward the inactive Meta-I state^{22,41}”*

24) p. 16, "results in motion of the retinal PSB proton away from the stabilizing interaction with its counter-ion Glu113" - a bit unorthodox, as normally people speak about the PSB counterion rather than proton counterion.

Response: The issue here is that the nitrogen of the PSB actually bears partial negative charge, while the positive charge on the retinal PSB contributed by protonation of this nitrogen is distributed on various atoms on the retinal and lysine side chain. The *largest* partial positive charge is on the Schiff base proton. Hence, the relative orientation of the N-H group of the PSB linkage relative to the Glu113 carboxyl group is what matters.

25) p. 17, "Upon isomerization, the C19 methyl group rotates in the opposite direction" - reference is needed

Response: References added.

Pages 18. "*Upon isomerization, the C19 methyl group rotates in the opposite direction (i.e. counter-clockwise) to the C20 methyl group*^{18,25,26}"

26) p. 21, "¹³C label rhodopsin" - style

Response: The sentence has been revised.

Page 19. "¹³C labeled rhodopsin"

27) Fig. 1 legend - PDB accession should be given; positions of the diagonal resonances used should be indicated; "scaled to the C12-C20 cross peaks" - specify how;

Response: The figure legend has been revised.

(b) "Crystal structure of rhodopsin (PDB ID 1U19) showing interactions of the C18, C19, C20 retinal methyl groups with surrounding residues."

"The rows selected correspond to the diagonal chemical shifts of the ¹³C18 resonance in rhodopsin at 21.6 ppm and Meta-II at 20.9 ppm, which maximize the crosspeak intensities.

Rows through the ¹³C19 diagonal resonances in rhodopsin (black) at 14.7 ppm and Meta-II (red) at 13.8 ppm obtained with the receptor regenerated with ¹³C8, ¹³C19 retinal and incorporating ¹³C-ring Phe and ¹³C ζ -labeled Tyr.

Rows through the ¹³C20 diagonal resonances in rhodopsin (black) at 16.4 ppm and Meta-II (red) at 13.7 ppm obtained with the receptor regenerated with ¹³C12, ¹³C20 retinal and incorporating ¹³C ζ -labeled Tyr.

Rows are taken through the ¹³C ζ -Tyr diagonal resonance at 155.2 ppm in rhodopsin and 156.1 ppm in Meta II."

We now describe the scaling of C12 and C20 cross peaks in Supplementary Fig. 2.

"Strong cross peaks are also observed between the ¹³C20 retinal resonance and the ¹³C ζ -resonance of Tyr268. The weak cross peak between ¹³C12 and ¹³C ζ -Tyr268 (relative the C12-C20 "internal control") indicates a longer internuclear distance. The intensity of the C12-C20 "internal control" allows us to scale the ¹³C12-¹³C ζ -Tyr268 and the ¹³C20 - ¹³C ζ -Tyr268 cross peaks relative to each other."

28) Fig. 1H - Signal-to-noise is poor resulting in some negative bands (phasing?). May be comment on that?

Response: We re-examined the spectra and the “negative peak” is just an artifact of poor signal-to-noise. We have other data in which there is not a negative peak, but these data were obtained by scanning rows to get the maximum C14 and C15 cross peak intensity are the best. The data sets on rhodopsin and Meta II were obtained and scaled to be comparable.

29) Fig. 3 legend - "are shown of rhodopsin minus Meta-I (orange) and rhodopsin minus Meta-II (black) of rhodopsin" - style

Response: The sentence has been revised.

Fig. 3 legend, Page 30. *“One-dimensional NMR difference spectra are shown of rhodopsin minus Meta-I (orange) and rhodopsin minus Meta-II (black) using rhodopsin containing labeled $^{13}\text{C}_\epsilon\text{-Met}$ (a), $^{13}\text{C}_\beta\text{-Ser}$ (b) or $^{13}\text{C}_\xi\text{-Tyr}$ (c).”*

30) Fig. 3 - large perturbations of Met and Tyr other than those discussed are observed, but ignored. Assignments of those resonances are not explained (reference if from the previous work)

Response: References have been added.

31) Fig. 4 legend refers to methods for the reference FTIR spectra, but those are not described there.

Response: Corrected. We have expanded the description of the FTIR methods under Methods and added the reference spectra in Supplementary Fig. 7.

32) Fig. 3 - two panels are labeled C

Response: Corrected.

33) Suppl. p. 4 - "the $^{13}\text{C}_\epsilon\text{-Met}$ "

Response: Corrected.

34) Suppl. p. 5 - for the discussion of the ring inversion and other possible changes in retinal (resulting in C18-Phe261 proximity) a figure would be nice

Response: We have now included schematics in Supplementary Figure 1.

35) Suppl. Fig. S4 legend and the following paragraph overlap significantly. May be merge the two.

Response: Merged. (This was an error.)

36) Suppl. p. 9 - two references to non-existing Fig. 3E are made.

Response: Corrected. The paragraph in point 35 above merged. The discussion was duplicated. (This was an error.)

37) Suppl. p. 12 - "absence of a cross peak between the retinal chromophore and $^{13}\text{C}\zeta$ -labeled tyrosine in Meta-II" - not true in general, should specifically name the retinal part discussed

Response: Corrected.

Page 8 (now Supplementary Fig. 4). *"This conclusion was based in part on the absence of a cross peak between the $^{13}\text{C}18$ methyl group on the β -ionone ring of the retinal chromophore and $^{13}\text{C}\zeta$ -labeled tyrosine in Meta-II (Fig. 1c)."*

38) Suppl. Fig. S6 - normalization of the C5-C18 cross peak is confusing, as its amplitude decreases

Response: We have revised the text to better convey the fact that the C18-Tyr intensity is increasing relative to the decaying Meta II signal from the C5-C18 cross peak.

Page 9 (now Supplementary Fig. 4). *"Panel (d) shows the row through the C18 diagonal in Meta II showing the intense C5-C18 cross peak (black trace). There is very little intensity to tyrosine resonances. Comparison of the same row after the decay of Meta II (purple trace) shows a strong increase in the C18-tyrosine cross peaks. The spectra are normalized to the C5-C18 cross peak to emphasize the increase in the C18-Tyr cross peak relative to the C5-C18 cross peak as Meta II decays. The largest increases in tyrosine intensity are associated with a $^{13}\text{C}18$ methyl resonance that has shifted to ~ 22.1 ppm. From the 2D plot in panel (b), one can see that the tyrosine resonances are associated with a C18 resonance at higher frequency."*

39) Suppl. p. 13 - (S.O. Smith, personal communication) - not sure one can refer to personal communication from one of the authors. May be unpublished is more appropriate.

Response: Corrected.

40) Suppl. p. 13 - "The high frequency of the $^{13}\text{C}5$ chemical shift suggests that the retinal with the flipped ring orientation is bound to Lys296 as a protonated Schiff's base" - explain and reference

Response: We have expanded and referenced the discussion concerning this point.

Page 9 (now Supplementary Fig. 4). *"The high frequency of the $^{13}\text{C}5$ chemical shift is similar to that in rhodopsin at 131.0 ppm and suggests that the retinal with the flipped ring orientation is bound to Lys296^{7,43} as a protonated Schiff's base. That is, the ^{13}C chemical shifts of the odd numbered carbons of the retinal polyene chain are sensitive to electron delocalization along the chain and are generally higher in frequency (downfield chemical shift) in protonated retinal Schiff bases compared to unprotonated Schiff bases²²."*

41) Suppl. p. 14 - I would move some of the very interesting discussion on Meta III into the main text.

Response: We have now included the observation of an all-trans PSB chromophore (i.e. a Meta-III intermediate) following Meta-II in the Discussion section where we present the data showing that the β -ionone ring has flipped.

Page 16. *“The final conformation observed by NMR is possibly Meta-III, in which the Schiff base is reprotonated⁴². The observation of an all-trans retinal PSB following Meta-II suggests that the reprotonation event is the driving force for rotation along the long axis of the retinal.”*

42) Suppl. Fig. 7 - the legend needs tightening, it is not very clear and accessible.

Response: The Figure S7 caption (now Supplementary Fig. 8) has been revised extensively. We completely agree that the original version of this figure was not very comprehensible.

Reviewer 2

Reviewer 2 describes the manuscript as “a very informative, if not groundbreaking, study on vertebrate rhodopsin” and indicates that “overall, this is an innovative, technically sound study which provides important new insights into structural changes leading to rhodopsin activation.”

The reviewer lists several minor issues that need correction prior to publication.

1. Figure 1 caption is confusing: Does Figure 1B show crystal structure of dark state or Meta-II intermediate. Also cytoplasmic and exterior membrane surface should be labeled in Figure 1B. Note that Tyr191 does not appear to be the closest residue to retinal C18. If Figure 1B is the dark state then this is understandable but in that case one of the Meta II forms of the crystal structure should also be shown.

Response: Figure 1b is the dark-state crystal structure. We have now added the PDB ID in the figure legend. We have also labeled the two surfaces and revised the figure caption. We show the comparison with the Meta-II crystal structure in Supplementary Fig. 1. We believe the new figure in Supplementary Fig. 1 will help the reader in visualizing the differences between the retinal orientation derived from the NMR measurements and observed in the crystal structures.

2. Page 6: It should be noted that the referred to peak at 21.6 ppm is not shown.

Response: The 21.6 ppm resonance corresponds to the C18 diagonal resonance. This is not shown in the portion of the spectrum displayed. We have added a supporting figure (new Figure S2) that explains how where the rows and cross peaks come from in a full 2D solid-state NMR spectrum. This should help the reader that is not familiar with this type of data.

3. Figure 1C: In bottom row of DARR data add lines to show the Phe261-C18 cross peaks as done for DARR row above.

Response: Lines have been added as suggested.

4. Page 7, paragraph 2: Thr118 is not shown in Figure 1B but mentioned in text. The authors should consider adding this to Figure.

Response: Thr118 has been added to Figure 1b.

5. Figure 2B: The label Ile189 is partially clipped at top of font.

Response: We were not able to find the label that the reviewer is referring to. We have double-checked all of the figures to ensure that they are not being clipped.

6. Figure 3: Crystal structure is mislabeled C and not D.

Response: Corrected

7. Page 12, second paragraph: Refer to Figure 3D (now 3C).

Response: Corrected

8. Figure 4: It would be easier to compare these plots if the pKas (inflection points) were indicated on the curves. This is especially true since x-axis pH scale is not reproduced on top set of panels.

Response: We added the pH scale on the top set of panels and added one vertical dashed line in each panel at the pKa in order to guide the eye.

9. Discussion: An earlier study which involved FTIR difference spectroscopy of isotope labeled rhodopsin suggested a role for Tyr268 in rhodopsin activation as well as helix-6 reorientation. This work should be mentioned and referenced to in Discussion (DeLange, et al. (1998) Tyrosine structural changes detected during the photoactivation of rhodopsin. J Biol Chem 273, 23735-23739).

Response: We have added this reference and a short discussion of the work.

Page 17. *“These results are consistent with an earlier FTIR study suggesting a role for Tyr268 and H6 reorientation in rhodopsin activation⁴⁴. ”*

10. Supplementary Figure S5: Panel D is not described. For example, which state does purple colored structure refer to?

Response: Additional explanation is added to the figure legend.

Page 19 (now Supplementary Fig. 7). *“The purple cylinders are a cartoon representation of the mechanism proposed here in which deprotonation and the associated changes in the extracellular hydrogen-bonding network allow the intracellular end of H6 to pivot inward and the extracellular end of H6 to pivot outward.”*

Reviewer 3

The authors present an extensive solid-state NMR study supported by FTIR and mutagenesis on rhodopsin. The aim of their work is to show that the activation of rhodopsin requires multiple steps of interaction between the retinal chromophore and opsin after the isomerisation event. Recent X-ray studies have shown differently oriented retinal co-factors. Solid-state NMR is here the method of choice to obtain specific data to resolve this issue.

Isotope labeled retinal was incorporated into residue-selectively labelled opsin. The observation of dipolar contacts between the ^{13}C spins enabled the authors to conclude details on the retinal orientation within the binding pocket after light activation and trapping the Meta-II state.

The authors suggest that in the first stage of the activation, the retinal beta-ionone ring triggers rotation of helix 6 and stabilizes Meta II. In the second step, C19 and C20 in the retinal polyene chain are suggested to sterically interact with Tyr191 and Tyr268, which also includes deprotonation of the retinal Schiff base.

Overall, data interpretation is sound and solid-state NMR is certainly the right approach.

1. However, I could not follow how the authors derived a sequential step model, for which time-resolved data or at least trapping of batho, meta-I and meta-II would have been required.

Response: The experiments described indeed make use of low-temperature trapping of both Meta-I and Meta-II and investigation by both NMR and FTIR. In addition, we make extensive use of previous experimental results from both our laboratory and others. We have revised the text extensively to more clearly describe how the experimental data have been integrated to generate the proposed model.

Abstract. "We integrate these observations with previous structural and functional studies to propose a two-stage mechanism for rhodopsin activation to describe how absorbed light energy is channeled into the protein to direct the known outward rotation of transmembrane helix H6, a hallmark of all active G protein-coupled receptors."

Introduction (Pages 5). "In refining the orientation of the all-trans retinal in Meta II based on solid-state NMR measurements, we propose an activation mechanism that builds on previous studies by emphasizing the changes in extracellular residues in close proximity to the retinal. We are able to follow changes in these residues by comparing differences between rhodopsin and Meta-II, or in some cases, between rhodopsin and Meta-I, which precedes Meta II in the photoreaction pathway. To address how specific residues in close association with the retinal control the equilibrium between Meta-I and Meta-II (i.e. the final step in the reaction pathway), we take advantage of FTIR spectroscopy. FTIR difference spectroscopy provides a complementary approach to NMR for characterizing the contribution of hydrogen bonding interactions of specific amino acids to receptor activation. The mechanism that emerges from these studies implicates steric interactions as the dominant force driving structural changes between rhodopsin and Meta I, while electrostatic (and hydrogen bonding) interactions control the formation of the active Meta II conformation."

2. The materials and method section is very short. Some more details would be helpful, especially, regarding the state of the samples. Was solid-state NMR applied to frozen detergent

solution or to proteoliposome samples? What is the source organism of the rhodopsin used here?

Response: Methods section has been expanded. The samples were frozen in DDM detergent and the source organism was cows (bovine rhodopsin).

Page 19. ***“Expression and purification of ¹³C labeled bovine rhodopsin for solid-state NMR. Isotope enriched bovine opsin⁵² was expressed using inducible HEK293S cell lines. The original cell lines were obtained from Jeremy Nathans (Johns Hopkins University), but not authenticated or tested for mycoplasma contamination. HEK293S cells are widely used for production of recombinant proteins and viruses. The expressed opsin was generated into rhodopsin through incubation with ~30 micromolar 11-cis retinal, extracted from membranes using 1% (w/v) n-β-D dodecyl maltopyranoside (DDM) in PBS pH 7.4, and purified using Rho-1D4-Sepharose resin^{53,54}. ¹³C labeled retinal was prepared synthetically. For Meta I, the DDM of the solubilized rhodopsin was reduced to 0.02% w/v and subsequently exchanged for 0.02-0.05% w/v digitonin on the 1D4-Sepharose column as previously described³². Rhodopsin is eluted in 2 mM phosphate buffer (pH = 7.0) containing digitonin (0.02-0.05% w/v) and 100 mM C-terminal nonapeptide. The pooled, eluted rhodopsin fractions were concentrated to a final volume of ~400 μL using Centricon devices with a 10 kDa molecular weight cut-off (Amicon, Bedford, MA), followed by further concentration under argon gas to a volume of ~100 μL. All buffers were prepared fresh before purification³².”***

3. It would be helpful to include some of the 2D spectra leading to the data in Fig. 1. The authors only show 1D slices of the spectra.

Response: We added a supporting figure (Supplementary Fig. 2) showing the full 2D plot for the experiment using ¹³Cξ-Tyr and ¹³C12,20-retinal. This figure is intended to illustrate where the rows originate from in Figure 1 and where the contour plots showing originate from in Figure 2.

4. Fig. 1G: The authors mention a comparison to build-up curves from model compounds but it is not clear whether these data are shown here. Furthermore, spin-diffusion build-up curves can be difficult to quantify. The authors seem to present here calculated curves for certain distances. Further details should be provided.

Response: Added in Methods

Pages 20. ***“The build-up curves were obtained by collecting DARR NMR spectra as a function of the mixing time during which magnetization is exchanged. The known curves were derived from measurements within rhodopsin at fixed distances (for e.g. retinal C5-C18, 1.4 Å, C8-C19, C12-C20, 2.4 Å, Cys110 Cβ-Cys187 Cβ, 3.6 Å; Cys187 Cβ- Gly188 Cα, 4.6 Å; Cys187 Cβ- Gly188 C=O, 5.3 Å). Spin diffusion is limited by the sparse labeling schemes that are typically used. The NMR assignments were based on mutation and/or mapping out specific correlations with unique resonances that have previously been assigned^{16,17,29,32,56,57}.”***

5. Some explanations should be provided how the MI/MII trapping was performed and how good the trapping efficiency was.

Response: Added in Methods

Page 19-20. "NMR measurements were first made on rhodopsin in the dark at -83 °C. For Meta- I and -II, the NMR MAS rotor containing rhodopsin was ejected from the NMR probe, the NMR cap on the rotor was removed and the sample was illuminated for 1-2 min using a 400 W lamp with a 495 nm long-pass filter at room temperature (Meta-II)⁵⁵ or 4 °C (Meta-I)³². The cap was then replaced and the rotor inserted into a pre-cooled NMR probe where the sample temperature was able to reach -83 °C within ~5 min. We estimate that the conversion from rhodopsin to Meta-I is >90% on the basis of UV/vis absorption and NMR spectra³². After conversion, we estimate that there is a loss of <10% of the Meta-I intermediate to Meta-II and opsin before the sample is cooled to -83 °C for NMR measurements³². We estimate that the conversion from rhodopsin to Meta II is >90% and the loss of Meta II to opsin is <5% before the sample is cooled to -83 °C⁵⁵."

6. It should be somewhere summarized how the assignment was achieved.

Response: Added in Methods.

Page 20. "The NMR assignments were based on mutation and/or mapping out specific correlations with unique resonances that have previously been assigned^{16,17,29,32,56,57}."

Reviewer #1 (Remarks to the Author):

The revised manuscript was substantially improved by better connecting the experimental results with the two stage hypothesis and by putting them into the context of previous knowledge. I only have minor suggestions, mainly editorial in nature.

1) Several places in the manuscript (especially in the supporting file) are hard to read due to a very dense and at times rambling style. I suggest to try to streamline wherever is possible to make it more accessible and readable.

2) line 123, "the orientation of the β -ionone ring has rotated" - may be simply "the β -ionone ring has rotated"?

3) line 137/141, "Crosspeaks appear between the diagonal resonances" - may be simply "Crosspeaks appear between resonances"?

4) line 157 - punctuation

5) line 185 - "with rhodopsin ^{13}C -labeled at $^{13}\text{C}\zeta$ -tyrosine" - style, 13 is not needed second time

6) line 219 - "is different than captured in" - style

7) line 243-244 - spectra....allows

8) line 349 - "drive the outward rotation of TM helix H6" - confusing, as both rotation and tilt are discussed, and I suspect the authors meant tilt (pivoting), not rotation, here.

9) line 441 - "of this helix with Pro2676.50, the conserved proline on H6, serving as the pivot point" - style, "this helix" is not clear.

10) line 458 - Greek font problem

11) line 499 - "sandwich samples" - not everyone knows what this is, windows and spacers could be indicated

12) line 508 - "For FTIR, the analysis uses difference methods, which allows one" - not clear.

13) line 682 - "incorporating either ^{13}C -ring labeled Phe and $^{13}\text{C}\zeta$ -labeled Tyr" - grammar

14) supp p.3 - "using dark rhodopsin" - jargon

15) supp p. 5 - "sites that separated in space" - grammar

16) supp p. 7 - I still have a problem with the lack of graphics showing the position of Met 207

17) supp p. 7 - "ring is much lower in the retinal binding site" - not clear

18) supp p. 10 - Mahlingam (misspelled)

19) supp p. 10 - 2 sentences in the consecutive paragraphs are almost the same, please remove the redundancy - "The low temperature trapped Meta-II observed by NMR corresponds to a Meta-II substate that precedes the rotation of the retinal as observed by protein crystallography" AND "One possibility is that the Meta-II intermediate observed by NMR corresponds to the Meta-II conformation or substate that precedes the rotation of the retinal observed in the Meta-II-opsin and Meta-II-M257Y crystal structures"

20) supp p. 11 - "chemical shifts along the retinal SB in" - not clear

21) supp. p. 14 - "would serve shift the position" - style

22) supp p. 16 - is influence - grammar

Reviewer #2 (Remarks to the Author):

The authors have adequately addressed all of the points raised by this reviewer.

1. The revisions in Figure 1, its caption and supplementary Figure 1 now provide sufficient clarification for the reader to visualize differences the retinal orientation between the NMR and crystal derived structures.
2. The addition of the Figure S2 shows more clearly the origin of the rows and cross peaks for the full 2D solid-state NMR spectrum.
3. Lines have now been added to show the Phe261-C18 cross peaks.
4. Thr118 has been added to Figure 1B
5. Agree
- 6 and 7.-corrected.

I strongly recommend publication of this paper which will be of significant interest.

Reviewer #3 (Remarks to the Author):

The authors have carefully addressed all of my concerns.

I recommend publication.

Point-by-Point Response to Reviewers

Reviewer #1:

The revised manuscript was substantially improved by better connecting the experimental results with the two stage hypothesis and by putting them into the context of previous knowledge. I only have minor suggestions, mainly editorial in nature.

1) Several places in the manuscript (especially in the supporting file) are hard to read due to a very dense and at times rambling style. I suggest to try to streamline wherever is possible to make it more accessible and readable.

Response: The Supplementary Figure legends have all been revised and shortened. To conform to the format of Nature Communications, we have moved much of the text in the Supplementary Figures to Supplementary Notes. In addition, to deleting non-essential text, this change has made the Supplementary material more accessible and readable.

2) line 123, "the orientation of the β -ionone ring has rotated" - may be simply "the β -ionone ring has rotated"?

Response: Corrected

3) line 137/141, "Crosspeaks appear between the diagonal resonances" - may be simply "Crosspeaks appear between resonances"?

Response: Corrected

4) line 157 - punctuation

Response: Corrected

5) line 185 - "with rhodopsin ^{13}C -labeled at $^{13}\text{C}\zeta$ -tyrosine" - style, 13 is not needed second time

Response: Corrected

6) line 219 - "is different than captured in" - style

Response: Corrected

7) line 243-244 - spectra....allows

Response: Corrected

8) line 349 - "drive the outward rotation of TM helix H6" - confusing, as both rotation and tilt are discussed, and I suspect the authors meant tilt (pivoting), not rotation, here.

Response: Corrected

9) line 441 - "of this helix with Pro2676.50, the conserved proline on H6, serving as the pivot point" - style, "this helix" is not clear.

Response: Corrected

10) line 458 - Greek font problem

Response: Corrected

11) line 499 - "sandwich samples" - not everyone knows what this is, windows and spacers could be indicated

Response: Corrected. *"Samples were prepared by drying solutions of rhodopsin between two CaF₂ windows and then pre-equilibrating the sample with buffer (200 mM Bis-tris propane or MES at pH 5.0 and 5.5) at the appropriate pH."*

12) line 508 - "For FTIR, the analysis uses difference methods, which allows one" - not clear.

Response: Corrected. *"For FTIR, the analysis uses difference methods in which the FTIR spectrum of Meta-I or Meta-II is subtracted from the spectrum of rhodopsin. Only the vibrations that change in frequency or intensity contribute to the difference spectrum. Changes in pH or temperature can be used to shift the equilibrium between Meta-I and Meta-II."*

13) line 682 - "incorporating either ¹³C-ring labeled Phe and ¹³Cζ-labeled Tyr" - grammar

Response: Corrected. *"¹³C-ring labeled phenylalanine or ¹³Cζ-labeled tyrosine."*

14) supp p.3 - "using dark rhodopsin" - jargon

Response: Corrected. (There were a number of places where we described rhodopsin as dark rhodopsin or the dark-state of rhodopsin. We have now changed these to just rhodopsin.)

15) supp p. 5 - "sites that separated in space" - grammar

Response: Corrected. *"sites that are separated in space"*

16) supp p. 7 - I still have a problem with the lack of graphics showing the position of Met207

Response: Corrected. We have added a graphic in Supplementary Figure 3.

17) supp p. 7 - "ring is much lower in the retinal binding site" - not clear

Response: Corrected. *"However, the crosspeak between the ¹³C5 retinal resonance and the ¹³C=O resonance of His211^{5,46} is only consistent with MD simulations where the β-ionone ring is positioned toward the intracellular side of the retinal binding site (Supplementary Table 1)."*

18) supp p. 10 - Mahlingam (misspelled)

Response: Corrected. "Mahalingam"

19) supp p. 10 - 2 sentences in the consecutive paragraphs are almost the same, please remove the redundancy - "The low temperature trapped Meta-II observed by NMR corresponds to a Meta-II substate that precedes the rotation of the retinal as observed by protein crystallography" AND "One possibility is that the Meta-II intermediate observed by NMR

corresponds to the Meta-II conformation or substate that precedes the rotation of the retinal observed in the Meta-II-opsin and Meta-II-M257Y crystal structures"

Response: Corrected. Redundancy deleted.

20) supp p. 11 - "chemical shifts along the retinal SB in" - not clear

Response: Corrected. *"Comparison of the the retinal SB ¹³C chemical shifts in Meta-II with those measured for all-trans retinal SB model compounds in various solvents show that the chemical shifts (and hence the structure and/or environment) of the retinal SB in Meta-II are unusual²⁰."*

21) supp. p. 14 - "would serve shift the position" – style

Response: Corrected. *"This motion would serve to shift the position of Tyr268^{6.51}."*

22) supp p. 16 - is influence - grammar

Response: Corrected. *"is influenced by the position of"*

Reviewer #2

The authors have adequately addressed all of the points raised by this reviewer.

1. The revisions in Figure 1, its caption and supplementary Figure 1 now provide sufficient clarification for the reader to visualize differences the retinal orientation between the NMR and crystal derived structures.
2. The addition of the Figure S2 shows more clearly the origin of the rows and cross peaks for the full 2D solid-state NMR spectrum.
3. Lines have now been added to show the Phe261-C18 cross peaks.
4. Thr118 has been added to Figure 1B
5. Agree
- 6 and 7.-corrected.

I strongly recommend publication of this paper which will be of significant interest.

Reviewer #3

The authors have carefully addressed all of my concerns.~~AA~~
I recommend publication.

Reviewer #1 (Remarks to the Author)

The revised manuscript addressed all the remaining issues. The paper is interesting and strong, and the flow, style, and logic have been improved significantly.